# gLSTM: Mitigating Over-Squashing by Increasing Storage Capacity

**Hugh Blayney**[1]**, Álvaro Arroyo**[1]**, Xiaowen Dong**[1]**, Michael M. Bronstein**[1,2]

[1]University of Oxford
[2]AITHYRA
`hugh@robots.ox.ac.uk`

## Abstract

Graph Neural Networks (GNNs) leverage the graph structure to transmit information between nodes, typically through the message-passing mechanism. While these models have found a wide variety of applications, they are known to suffer from *over-squashing*, where information from a large receptive field of node representations is collapsed into a single fixed sized vector, resulting in an information bottleneck. In this paper, we re-examine the over-squashing phenomenon through the lens of model *storage and retrieval capacity*, which we define as the amount of information that can be stored in a node's representation for later use. We study some of the limitations of existing tasks used to measure over-squashing and introduce a new synthetic task to demonstrate that an information bottleneck can saturate this capacity. Furthermore, we adapt ideas from the sequence modeling literature on associative memories, fast weight programmers, and the xLSTM model to develop a novel GNN architecture with improved capacity. We demonstrate strong performance of this architecture both on our capacity synthetic task, as well as a range of real-world graph benchmarks.

## 1 Introduction

Graph Neural Networks (GNNs) (Sperduti, 1993; Gori et al., 2005; Scarselli et al., 2008; Micheli, 2009; Bruna et al., 2014; Defferrard et al., 2017) have emerged as a standard framework for learning on graph-structured data. The majority of these models follow a *message passing* paradigm, where nodes iteratively exchange information with neighbors, commonly referred to as Message-Passing Neural Networks (MPNNs). Examples of this family of architectures include GCN (Kipf & Welling, 2017), GAT (Veličković et al., 2018), GIN (Xu et al., 2018), and GraphSAGE (Hamilton et al., 2017).

Since each MPNN layer exchanges information between neighboring nodes to update node representations, the number of layers thus dictates the receptive field: the set of nodes over which information is aggregated. Deep MPNNs are, in theory, desirable as they can model long-range dependencies, but scaling to many layers has historically been difficult due to two pervasive issues that have received significant attention in the literature: *over-smoothing* and *over-squashing*. We focus on the latter in this work.

Over-squashing was initially identified by Alon & Yahav (2021) as a problem of compressing information from a node's receptive field into a fixed-size vector. This was linked with depth and long-range dependencies, since receptive fields tend to grow exponentially with depth. Later work (Topping et al., 2022; Di Giovanni et al., 2023a) identified that this bottleneck could also result in low sensitivity as measured by the node Jacobian, linking graph topology and aspects of model architecture via an upper bound on this Jacobian. This low sensitivity arises due to repeated degree normalization and application of a contractive nonlinearity over many layers. Arnaiz-Rodriguez & Errica (2025) suggest that these two descriptions of over-squashing are not the same, and that Alon & Yahav (2021) define it as a problem of computational graph bottlenecks, while later work often defines it as a problem of topological bottlenecks. We discuss this separation and its relation to our work in Appendix A.

Instead of separating by issues of computational tree structure and bottlenecks, we suggest an alternative separation by resultant *failure mode*: limited information storage capacity, and low sensitivity. In light of this, we highlight another divergence in the literature: the work of Alon & Yahav (2021) implicitly described over-squashing as a capacity problem, and later work re-framed it as a problem of sensitivity. Focusing on the failure modes themselves not only allows us to revisit the issue of capacity originally discussed in Alon & Yahav (2021), but also to more directly motivate benchmarks and remedies, both of which we discuss in this paper. We further contextualize this issue of capacity by studying it in *isolation* from sensitivity issues. We believe this focus not only provides a more complete understanding of over-squashing but also highlights new directions to mitigate it.

To combat over-squashing, existing research has focused on ameliorating topological bottlenecks through *rewiring* (Gasteiger et al., 2019; Gutteridge et al., 2023; Nguyen et al., 2023; Saber & Salehi-Abari, 2025) and controlling the flow of information (Bresson & Laurent, 2017; Finkelshtein et al., 2024; Errica et al., 2025) – targeting topological and general computational bottlenecks respectively. However, these bottlenecks are only an issue if they harm performance in some way: in this work we discuss issues of (1) reduced sensitivity and (2) saturated storage capacity. Our proposed architecture in Section 4 targets the latter failure mode: adapting the MPNN architecture to improve its *ability to store and retrieve information*. Framing over-squashing as a capacity limitation that can be addressed at the architecture level exposes a previously unexplored path, and our results validate this direction.

To improve MPNN storage capacity we turn to the sequence modeling literature, which has a long history of tackling equivalent problems (Hochreiter & Schmidhuber, 1997; Orvieto et al., 2023; Gu & Dao, 2023; Beck et al., 2024; Arora et al., 2024). Taking inspiration from these works, we introduce an MPNN architecture that utilizes associative memory (Beck et al., 2024; Schlag et al., 2021; Hopfield, 1982), and demonstrate that this exhibits improved storage capacity.

**Contributions.** Our main contributions are as follows. In Section 3, we **re-characterize over-squashing into two distinct failure modes**: *saturating capacity* and *low sensitivity*, which we term capacity over-squashing and sensitivity over-squashing respectively. We discuss in Section 3.1 the pitfalls of widely used over-squashing tasks, which either fail to evaluate capacity at all, or evaluate the two issues in tandem and are thus unable to separate their effects. In Section 3.2, we introduce a novel synthetic task, which to our knowledge is the first that **measures capacity over-squashing in isolation**. In Section 4, we present a new MPNN architecture based on the recent xLSTM architecture (Beck et al., 2024), which uses **associative memory to increase capacity**, explicitly targeting this capacity over-squashing viewpoint. Section 5 demonstrates that this architecture performs well on our synthetic capacity task and a range of real-world benchmarks, and Section 5.2 demonstrates empirically that **capacity over-squashing can occur separately from sensitivity over-squashing**.

## 2 BACKGROUND AND RELATED WORK

**Message Passing Neural Networks** Let a graph $\mathcal{G}$ be a tuple $(\mathcal{V}, \mathcal{E})$ where $\mathcal{V}$ is the set of nodes and $\mathcal{E}$ the set of edges. An edge from node $u$ to $v$ is denoted $(u, v) \in \mathcal{E}$. The connectivity is encoded by the adjacency matrix $\boldsymbol{A} \in \mathbb{R}^{|\mathcal{V}| \times |\mathcal{V}|}$, where $\boldsymbol{A}_{uv} = 1$ if $(u, v) \in \mathcal{E}$ and 0 otherwise. Each node $v$ has a feature vector $\boldsymbol{x}_v \in \mathbb{R}^d$.

GNNs are functions $f_{\boldsymbol{\theta}} : (\mathcal{G}, \{\boldsymbol{x}_v\}) \mapsto \boldsymbol{y}$ with parameters $\boldsymbol{\theta}$, trained via gradient descent to predict node- or graph-level labels $\boldsymbol{y}$. These models typically take the form of MPNNs, which compute latent representations by composing $L$ layers of the following node-wise operation:

$$\boldsymbol{h}_u^{(l)} = \phi^{(l)}\big(\boldsymbol{h}_u^{(l-1)}, \ \psi^{(l)}(\{\boldsymbol{h}_v^{(l-1)} : (u, v) \in \mathcal{E}\})\big), \tag{1}$$

where $\psi^{(l)}$ is a permutation-invariant *aggregator*, $\phi^{(l)}$ combines neighbor messages with the previous embedding and $\boldsymbol{h}_v^{(0)} = \boldsymbol{x}_v$. Throughout, we use "GNN" and "MPNN" interchangeably. Note we depart from the more usual notation of $k$ for layer index to avoid confusion with *keys*, introduced in Section 3.2. The most commonly used aggregation function takes the form

$$\psi^{(l)}(\{\boldsymbol{h}_v^{(l-1)} : (u, v) \in \mathcal{E}\}) = \sum_v \boldsymbol{O}_{uv} \, \boldsymbol{h}_v^{(l-1)}, \tag{2}$$

where $\boldsymbol{O} \in \mathbb{R}^{|\mathcal{V}| \times |\mathcal{V}|}$ is some message-passing matrix. For GCN (Kipf & Welling, 2017), $\boldsymbol{O} = \tilde{\boldsymbol{D}}^{-1/2} \tilde{\boldsymbol{A}} \tilde{\boldsymbol{D}}^{-1/2}$ with $\tilde{\boldsymbol{A}} = \boldsymbol{A} + \boldsymbol{I}$ for diagonal $\tilde{\boldsymbol{D}} \in \mathbb{R}^{|\mathcal{V}| \times |\mathcal{V}|}$ with $\tilde{\boldsymbol{D}}_{ii} = \sum_j \tilde{\boldsymbol{A}}_{ij}$. We frequently

denote the set of message-passing neighbors of node $u$ as $\mathcal{N}_u = \{v \in \mathcal{V} \mid \boldsymbol{O}_{uv} \neq 0\}$ – if the message-passing matrix is layer-dependent, we may superscript this with a layer index.

**Fast Weight Programmers**   Fast Weight Programmers (FWPs) are a class of neural network motivated by the idea of allowing variable network weights dependent on the input - termed *fast weights*. One method to "program" the fast weights is to take outer products of learned projections of the input (Schmidhuber, 1992). Schlag et al. (2021) observe that – up to normalization and activation function differences – linear Transformers (Katharopoulos et al., 2020) are equivalent to FWPs.

**xLSTM: Associative Memory for Language Modeling**   Recent work (Beck et al., 2024; 2025) introduced xLSTM, a development of the original LSTM (Hochreiter & Schmidhuber, 1997) architecture that resulted in a performant recurrent neural network capable of language modeling. Of relevance to our work are the following limitations that xLSTM aims to address: the inability to "revise storage decisions" and the limited storage capacity of the scalar cell states. The first of these is addressed through modifying the original LSTM gating to use exponential activation functions. The second is addressed by introducing associative memory, updated using an outer product update rule equivalent to that of FWPs to store keys and values (see Appendix B for more details).

## 3   THE TWO FAILURE MODES OF OVER-SQUASHING

Over-squashing was initially introduced by Alon & Yahav (2021) as an issue of **storage capacity**. They observed that recurrent sequence models exhibit a bottleneck in representing all the information from their past inputs, and this bottleneck exists in a more harmful form in GNNs, in which the information receptive field grows exponentially. They introduced a synthetic task to measure over-squashing by propagating information through various sizes of binary tree.

Later research identified that this computational graph bottleneck *also* resulted in **low sensitivity** and issues of signal propagation. Topping et al. (2022); Di Giovanni et al. (2023a) quantified this low sensitivity via the Jacobian of node representations, establishing the following sensitivity bound: for an MPNN with $l$ layers, $c_\sigma$ Lipschitz constant of the activation, $w$ maximal entry-value over weight matrices, $d$ embedding dimension and $u, v \in \mathcal{V}$, one has

$$\left\| \frac{\partial \boldsymbol{h}_v^{(l)}}{\partial \boldsymbol{h}_u^{(0)}} \right\|_{L_1} \leq \underbrace{(c_\sigma w d)^l}_{\text{model}} \overbrace{\left(\boldsymbol{O}^l\right)_{uv}}^{\text{topology}}, \tag{3}$$

where $\boldsymbol{O}$ is the message passing matrix used by the MPNN as in Equation (2). This bound establishes that low sensitivity results from both graph topology as well as factors intrinsic to the MPNN model. In particular, sensitivity is lowered by the nature of the message-passing, where the culprit is successive powers of a degree-normalized adjacency matrix. It is also lowered by the contractive nature of the nonlinearity $\sigma$ and the values of the weight matrices, as established in Arroyo et al. (2025). Despite this analysis being purely one of sensitivity rather than capacity, it was also termed over-squashing, and has been successful in establishing links to other areas, including the expressive power of MPNNs (Di Giovanni et al., 2023b) and graph effective resistance (Black et al., 2023).

We argue that there are two *distinct* problems arising from bottlenecks in MPNNs: *reduced sensitivity* (sensitivity over-squashing) and *saturating storage capacity* (capacity over-squashing). Due to the influential paper of Topping et al. (2022) the sensitivity viewpoint on over-squashing has thus far been the predominant approach in the literature; **in this work, we seek to revisit the storage capacity viewpoint** and investigate how this issue can be avoided. We define storage capacity as the amount of information that can be stored in a node's representation for later use: a representation is *saturated* when it is unable to store any more information.

**Conflation With Depth**   The vast majority of existing research links over-squashing with depth. To an extent, this is justified: the bound of Equation (3) decreases exponentially with MPNN depth, and real-world graphs tend to exhibit receptive fields that grow exponentially in depth, leading to capacity quickly becoming a problem for deep MPNNs. However, alongside recent work (Arnaiz-Rodriguez & Errica, 2025), we highlight that over-squashing is not *exclusively* a problem of depth: bottlenecks can be observed in single-layer GNNs acting on high-degree nodes – we exploit this fact in our synthetic task of Section 3.2. Furthermore, in studying over-squashing only in the *deep regime*, much of the literature has conflated the problem with issues of *vanishing gradients*, themselves closely linked to the related problem of *over-smoothing* (Di Giovanni et al., 2023b).

Arroyo et al. (2025) give a more precise treatment of how the issue of over-squashing relates to depth, through over-smoothing (zero collapse) and vanishing gradients. In this work we study over-squashing in the *shallow regime*: this allows us to isolate the issue of saturating capacity, avoiding the effects of depth on both reduced sensitivity (Equation (3)) and vanishing gradients.

### 3.1 EXISTING OVER-SQUASHING TASKS DO NOT (ONLY) TEST CAPACITY

An instructive way of contrasting sensitivity against capacity is via synthetic tasks. The most common of these used to assess over-squashing are the Ring-Transfer tasks of Di Giovanni et al. (2023a). The goal of these tests is for a MPNN to 'transfer' features contained at a target node to a source node, across a large graph distance. Various graphs are tested, in particular a ring of nodes, but the common feature is that there exists a long shortest-path from the source to target node. All of these exhibit an exponentially growing receptive field of at least $2^k$ for $k$ layers, since each node is connected to at least two others; repeated aggregation and application of MPNN layers and nonlinearities makes this a good test of the sensitivity-based view of over-squashing.

However, this task is particularly ill-equipped to test the issue of storage capacity, as the *only* relevant information in the graph is that of the target node, and all intermediate nodes are assigned constant vectors of ones. In this way, there is only a single node's representation worth of information to be transferred. It is unclear how much this task measures behavior found in real-world tasks: exponentially growing receptive fields will not be padded by nodes with identical representations. Figure 1 (left) visualizes the computational graph of RingTransfer, demonstrating that it is dominated by nodes containing no information. Therefore, this task exhibits a large computational bottleneck without any issues of saturating capacity: this highlights the fact that, beyond the computational bottleneck, saturated capacity is at least also dependent on the information content of the task.

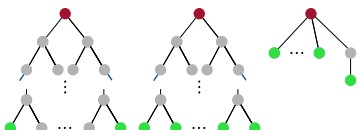

Figure 1: Computational graphs. **Left**: Ring-Transfer (Di Giovanni et al., 2023a). **Middle**: `Tree-NeighborsMatch` (Alon & Yahav, 2021). **Right**: NAR, introduced in Section 3.2. Nodes with informative features are green, background gray. Red node is trained to solve the task.

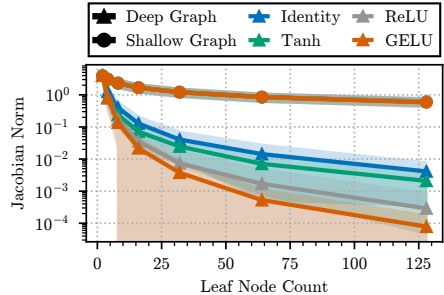

Figure 2: Log Jacobian norms. "Deep" graphs are binary trees of `Tree-NeighborsMatch` (Alon & Yahav, 2021); "Shallow" graphs are single-level trees with the same number of leaves. A GCN of depth equal to the tree depth acts on each. Jacobian norms are $|\partial \boldsymbol{h}_r^{(L)}/\partial \boldsymbol{h}_l^{(0)}|_{L_1}$ for root $r$ and leaf $l$ (red/green in Figure 1). Shaded area is standard deviation.

Alon & Yahav (2021) introduced the `Tree-NeighborsMatch` task to measure capacity by propagating information from the leaf nodes of a variable-size binary tree. It shares similarities with the task we introduce in Section 3.2 in that it controls the amount of information that is forced into a single node representation. However, it propagates this information through a deep binary tree, requiring variable-depth MPNNs. This significantly harms sensitivity: we visualize Jacobian norms of a GCN acting on a deep binary tree vs a single layer tree with matching leaf counts in Figure 2, demonstrating that this sensitivity drops off far faster for deep GCNs. This is unsurprising given the bound of Equation (3): deep GCNs must additionally contend with "model" squashing terms of nonlinearity and weight contraction that scale exponentially with depth. Therefore performance degradation trends are due to both 1) saturating capacity and 2) low sensitivity; deep tasks such as `Tree-NeighborsMatch` are impacted by both over-squashing issues, rather than isolating the issue of capacity.

### 3.2 NEIGHBOR ASSOCIATIVE RECALL: ISOLATING STORAGE CAPACITY

We investigate storage capacity by measuring *associative recall*: this is a common approach taken in the sequence-modeling literature (Ba et al., 2016; Schlag et al., 2021; Arora et al., 2024; Jelassi et al., 2024), in which the question of model storage capacity is also clearly of interest. These

synthetic tasks involve presenting the model with a sequence of key value pairs followed by a query that corresponds to one of the presented keys, and the model must return the associated value.

To this end, we introduce a task that we refer to as Neighbor Associative Recall (NAR). Whereas the sequence associative recall tasks measure the ability of a model to recall previous information from a variable-length sequence, our graph adaptation is designed to measure the ability of a GNN to recall information from the previous message passing round over a variable number of neighbors.

The task is designed as follows. For a given neighborhood size $N$ we create a graph of $N + 3$ nodes. This graph consists of $N$ "neighbor" nodes, a central node to which they are all connected, an intermediate node connected to the central node, and a "query" node connected only to the intermediate node. An example such graph is visualized in Figure 3.

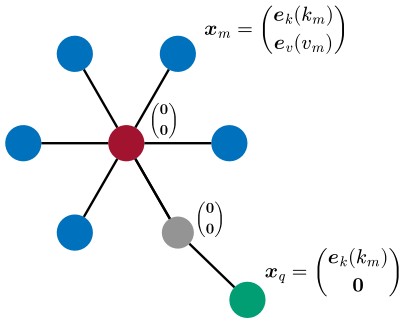

For a fixed neighborhood size $N$ we define a fixed set of keys and values, $N = |\mathbb{K}| = |\mathbb{V}|$, and a pair of learned vector embedding functions $\boldsymbol{e}_k : \mathbb{K} \to \mathbb{R}^{d_{\text{emb}}}$, $\boldsymbol{e}_v : \mathbb{V} \to \mathbb{R}^{d_{\text{emb}}}$ for embedding dimension $d_{\text{emb}}$. Each of the neighbor nodes $n$ has a different assigned key $k_n \in \mathbb{K}$, and also a value $v_n \in \mathbb{V}$, randomly sampled with replacement. The input feature vector of these nodes is a concatenation of the two learned embeddings $\boldsymbol{x}_n = [\boldsymbol{e}_k(k_n); \boldsymbol{e}_v(v_n)] \in \mathbb{R}^{2d_{\text{emb}}}$. The intermediate node and central node both have zero-valued feature vectors. Associated with the query node $q$ is a randomly sampled key-value node $m$; the input feature vector for the query node consists of the corresponding key embedding concatenated with a vector of zeroes, $\boldsymbol{x}_q = [\boldsymbol{e}_k(k_m); \boldsymbol{0}] \in \mathbb{R}^{2d_{\text{emb}}}$. The model is trained such that the central node must predict the value $v_m$ associated with the sampled key node. Training is via cross-entropy loss where the target of the central node is a one-hot vector corresponding to a fixed ordering of $\mathbb{V}$. This approach can be viewed as a graph adaptation of the sequence associative recall task of Schlag et al. (2021). Results are presented in Section 5.1.

Figure 3: An example graph with $N = 5$ from the NAR task. Key-value nodes are shown in blue, the central node in red and the query node in green. In this graph, $m$ is the randomly sampled index of the key-value node associated with query node $q$. The target for this graph is a one-hot vector corresponding to $v_m$.

An alternative formulation of this task with a regression target is discussed in Appendix D.8.

NAR is designed such that the receptive field of the central node will comprise *only* the neighbor nodes in the first layer. In order to perfectly solve the task, it must store all of the key-value information in this initial receptive field, as it is impossible to limit the scope of the information that might later be required. In the second layer, the receptive field will include the query node: now, the model must selectively recall the correct value from its immediate neighbors.

This task is novel as it assesses over-squashing in the shallow regime: MPNNs tested in Section 5.1 consist of just two message passing layers. This more effectively isolates the issue of capacity, without secondary effects from low sensitivity and vanishing gradients as visualized in Figure 2.

## 4 GLSTM: COMBINING GRAPH NETWORKS AND ASSOCIATIVE MEMORY

Prior work on over-squashing has focused almost exclusively on mitigating sensitivity issues, often through graph rewiring (Gasteiger et al., 2019; Gutteridge et al., 2023; Nguyen et al., 2023). Some work has implicitly tackled capacity over-squashing by moderating the flow of information into node representations (Bresson & Laurent, 2017; Finkelshtein et al., 2024; Errica et al., 2025) thus reducing capacity requirements, but we are unaware of any work that has attempted to *increase* capacity at an architecture level. Motivated by memory-capacity gains in sequence models (Ba et al., 2016; Beck et al., 2024), we introduce associative memory into an MPNN architecture to explicitly enlarge its information-storage capacity; we measure this in the graph setting using the NAR task introduced above. We further introduce the gating scheme of Beck et al. (2024) to investigate its efficacy in the graph setting, given strong sequence modeling performance. Since these adaptations are inspired in part by their successful use in xLSTM, we refer to our related graph architecture as gLSTM.

For any node $u$ at layer $l$, in addition to the usual MPNN vector hidden state $\boldsymbol{h}_u^{(l)}$, gLSTM maintains a matrix hidden state $\boldsymbol{C}_u^{(l)}$. The initial hidden state $\boldsymbol{h}_u^{(0)}$ is the input node feature vector $\boldsymbol{x}_u$. Keys and values are used to update $\boldsymbol{C}_u^{(l)}$ via an FWP-style outer product rule: these are projections of the previous vector hidden state $\boldsymbol{h}_u^{(l-1)}$. The next vector hidden state $\boldsymbol{h}_u^{(l)}$ is determined by "querying" $\boldsymbol{C}_u^{(l)}$ via matrix multiplication with another projection of the previous vector hidden states.

The modified gLSTM update equations are given below. Highlighted in blue are the differences to xLSTM. Biases correspond exactly to xLSTM (Appendix B) and are omitted for clarity.

**State (and normalization) updates:**

$$\boldsymbol{C}_u^{(l)} = f_u^{(l)} \boldsymbol{C}_u^{(l-1)} + \sum_{v \in \mathcal{N}_u^{(l)} \cup \{u\}} i_v^{(l)} \boldsymbol{v}_v^{(l)} \otimes \boldsymbol{k}_v^{(l)}$$

$$\boldsymbol{n}_u^{(l)} = f_u^{(l)} \boldsymbol{n}_u^{(l-1)} + \sum_{v \in \mathcal{N}_u^{(l)} \cup \{u\}} i_v^{(l)} \boldsymbol{k}_v^{(l)}$$

$$m_u^{(l)} = \max\left(\left\{\tilde{f}_u^{(l)} + m_u^{(l-1)}\right\} \cup \left\{\tilde{i}_v^{(l)} \mid \forall v \in \mathcal{N}_u^{(l)} \cup \{u\}\right\}\right)$$

**Query / Key / Value computation:**

$$\boldsymbol{q}_u^{(l)} = \boldsymbol{W}_q \left[\boldsymbol{h}_u^{(l-1)}; \sum_{v \in \mathcal{N}_u^{(l)}} \boldsymbol{h}_v^{(l-1)}\right]$$

$$\boldsymbol{k}_u^{(l)} = \frac{1}{\sqrt{d}} \boldsymbol{W}_k \boldsymbol{h}_u^{(l-1)}$$

$$\boldsymbol{v}_u^{(l)} = \boldsymbol{W}_v \boldsymbol{h}_u^{(l-1)}$$

The square brackets above denote vector concatenation. Concatenating the hidden state for the node and its neighbours in this way keeps them separate and allows the query – which will determine what is retrieved from the matrix memory – to *separately* depend on both the previous state of the node itself and the previous states of its neighbours.

**Gate computation:**

$$i_u^{(l)} = \exp\left(\tilde{i}_u^{(l)} - m_u^{(l)}\right) \qquad \tilde{i}_u^{(l)} = \boldsymbol{w}_i^T \boldsymbol{h}_u^{(l-1)}$$

$$f_u^{(l)} = \exp\left(\tilde{f}_u^{(l)} + m_u^{(l-1)} - m_u^{(l)}\right) \quad \tilde{f}_u^{(l)} = \boldsymbol{w}_f^T \boldsymbol{h}_u^{(l-1)}$$

$$\boldsymbol{o}_u^{(l)} = \sigma\left(\tilde{\boldsymbol{o}}_u^{(l)}\right) \qquad \tilde{\boldsymbol{o}}_u^{(l)} = \boldsymbol{W}_o \boldsymbol{h}_u^{(l-1)}$$

**Output:**

$$\tilde{\boldsymbol{h}}_u^{(l)} = \frac{\boldsymbol{C}_u^{(l)} \boldsymbol{q}_u^{(l)}}{\max\left\{\left|\boldsymbol{n}_u^{(l)\top} \boldsymbol{q}_u^{(l)}\right|, 1\right\}}$$

$$\boldsymbol{h}_u^{(l)} = \boldsymbol{o}_u^{(l)} \odot \tilde{\boldsymbol{h}}_u^{(l)}$$

**Block Structure**   Arroyo et al. (2025) note that sensitivity over-squashing issues are largely caused by vanishing gradients – a phenomenon well-explored in the sequence-modeling literature. In an attempt to address this, gLSTM uses a similar block structure to the mLSTM block upon which it is based. Of particular importance is the residual connection – which brings the norm of the layer-wise Jacobian to the edge of chaos – and use of input and hidden norms, which regulate the magnitude of the Jacobian norms. Figure 4 visualizes the block structure of gLSTM that we employ.

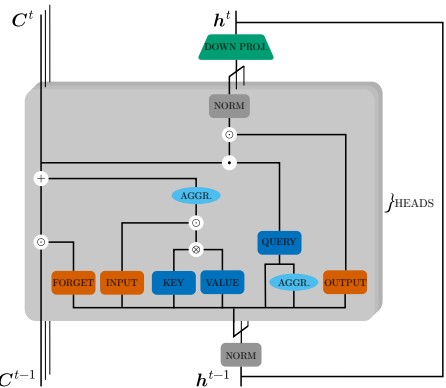

Figure 4: gLSTM block structure. Gates shown in orange, query/key/value in dark blue. *Aggr.* represents aggregation across neighborhoods. Symbols $\odot, \otimes, +, \cdot$ denote Hadamard product, outer product, vector addition, matrix multiplication.

**K-Hop Aggregation**   Following Arroyo et al. (2025) we combine the memory capabilities of the xLSTM block with a *highly connected* message passing graph structure: employing a k-hop aggregation scheme. In this setting, each node $u$ at layer $l$ will aggregate information from the neighborhood

$$\mathcal{N}_u^{(l)} = \{v \in \mathcal{V} \mid d_{\mathcal{G}}(u, v) = l\},$$

where $d_{\mathcal{G}} : \mathcal{V} \times \mathcal{V} \rightarrow \mathbb{R}_{\geq 0}$ is the length of the minimal walk connecting nodes $u$ and $v$. This approach resembles that of Ding et al. (2024), but with an additional recurrence: hidden states are used as input at each step. This substantially changes the way information can propagate through the graph. Furthermore, it also has links to ChebNet (Defferrard et al., 2017), which has recently been found to perform strongly on long-range tasks (Hariri et al., 2025).

This aggregation scheme appears to greatly improve gLSTM performance: our synthetic task in Section 5.1 significantly benefits from this aggregation scheme, and the ablations in Appendix D.2 demonstrate that it improves performance in all but one of the tested benchmarks. We hypothesize that – in addition to providing a highly connected computational graph that lessens over-squashing sensitivity bottleneck issues – this is because it also provides an extremely useful inductive bias for the *recall* mechanism of gLSTM. Information that has previously been stored in the associative memory is not then included in later message passing rounds, and later nodes are able to query this memory in isolation.

# 5 EXPERIMENTS

## 5.1 NEIGHBOR ASSOCIATIVE RECALL

We train various models on NAR with varying neighbor count $N$, with results shown in Figure 5a. Throughout this section we compare gLSTM using K-hop aggregation to GCN using standard aggregation, since gLSTM performs significantly better in this task when using K-hop aggregation whereas GCN performance is harmed by K-hop. We present additional results in Appendix D.7 where we separate by aggregation method and include results for a larger number of models. A comparison of the number of trainable parameters is shown in Figure 5b. Fair comparison between matrix and vector memory is nontrivial, so we select these parameter counts to "favor" GCN.

These results demonstrate that gLSTM shows significantly improved recall abilities compared to GCN. gLSTM retains perfect recall until the number of neighbors equals the memory dimension of the model: beyond this is where capacity over-squashing appears to become a problem. This agrees with intuition, since the maximum number of orthogonal key vectors (and separately, value vectors) is equal to the memory dimension. However, it is interesting to note how the performance decreases slowly as the neighbor count exceeds this limit, particularly for higher memory dimensions: this appears to be a graph analog of the "graceful saturation" described by Smolensky (1990). By contrast, capacity over-squashing starts much earlier at just $N = 8$ for the largest GCN model tested.

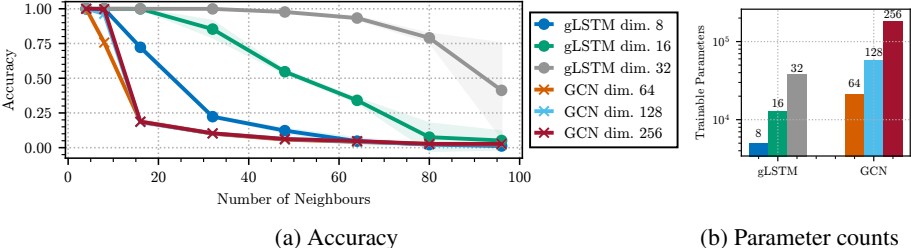

(a) Accuracy                    (b) Parameter counts

Figure 5: Test-set mean Accuracy (standard deviation shaded) for the NAR task, for gLSTM and GCN models with various hidden dimensions shown in Figure 5a, number of trainable parameters in Figure 5b. Note that gLSTM uses K-hop aggregation here, whereas GCN does not; see Appendix D.7 for separated performance by aggregation strategy.

## 5.2 HOW DOES CAPACITY RELATE TO SENSITIVITY?

In this section, we investigate empirically how capacity over-squashing – as measured by performance on NAR – relates to sensitivity over-squashing.

We directly measure the Jacobian norm of Topping et al. (2022); Di Giovanni et al. (2023a), computing the sensitivity of the output feature vector on the central (output) node $c$ to the input vectors on the key-value neighbor nodes $n$, $|\partial \boldsymbol{h}_c^{(2)}/\partial \boldsymbol{x}_n|_{L_1}$. These results are visualized in Figure 6a.

We see therefore that sensitivity, as measured by the Jacobian norm, does not correlate with NAR performance. Given that NAR performance degradation is due to capacity over-squashing, we therefore observe that **capacity over-squashing can occur without sensitivity over-squashing**. This is clear from the fact that 1) sensitivity increases consistently for GCN models above $N = 16$ to the point where it matches initial sensitivity, despite no increase in performance and 2) sensitivity for

gLSTM tends to carry on increasing beyond where performance starts to degrade. We note these trends – as with all observations we make in this section – hold true for the NAR regression task in Appendix D.8.1.

However, if we examine the difference in Jacobian norms between the neighbor nodes which are *selected* (those which have a key corresponding to the query node) vs *background*, we see trends that align with our notion of capacity. Figure 6a visualizes the ratio of Jacobian norms for selected nodes to that for background nodes. We observe that for all GCN models this ratio quickly falls to unity at the point where capacity over-squashing starts to occur, and gLSTM ratios consistently plateau – and start to slowly decrease – at their memory dimension, similarly coinciding with capacity over-squashing. It appears therefore that capacity over-squashing harms a model's ability to be selectively sensitive to different nodes in the NAR task.

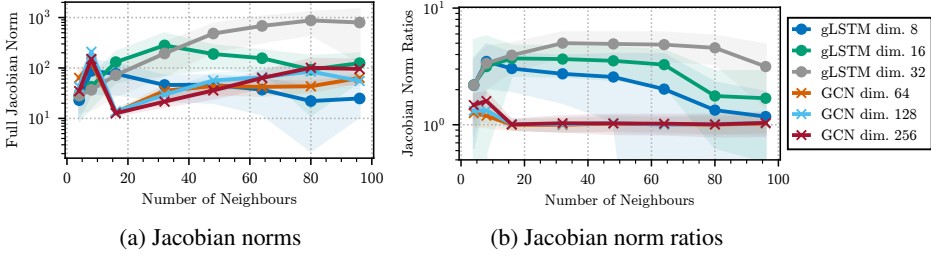

(a) Jacobian norms

(b) Jacobian norm ratios

Figure 6: **Left**: Mean Jacobian norms for different gLSTM and GCN models, with varying number of neighbors in the NAR task. **Right**: Mean ratio between the Jacobian norms of the selected (key corresponds to query) to background (key is different from query) neighbor nodes, for varying model dimensions. Standard deviation shaded in both plots.

Another over-squashing sensitivity metric is that of Di Giovanni et al. (2023b), who introduce the *maximal mixing* metric. For node-level function $\boldsymbol{Y} : \mathbb{R}^{n \times d} \to \mathbb{R}^{n \times d}$, the mixing of features associated with nodes $u, v$ at a given node $i$ is defined as

$$\operatorname*{mix}_{\boldsymbol{Y}}(i, v, u) = \max_{\boldsymbol{X}} \left\| \frac{\partial^2 \left( \boldsymbol{Y} \left( \boldsymbol{X} \right) \right)_i}{\partial \boldsymbol{x}_u \partial \boldsymbol{x}_v} \right\|.$$

Although motivated through intuition of mixing, we observe the mixed partial derivative can equally be viewed as a composition of partial derivatives quantifying *selective sensitivity* - how much the sensitivity with respect to one node feature varies with respect to another node feature. In this respect, we expect it to be highly relevant to the sensitivity ratios visible in Figures 6b and 10.

To study this empirically for NAR, we take the maximum over the measured Hessians for different models. These Hessian 3-tensors are large, so we further limit to a subset of the overall tensor in order to compute them on available hardware: we are most interested in how the output sensitivity to the neighbor *value* vectors varies with the *query* vector, so we limit to the corresponding input dimensions. For the central, neighbor and query nodes $c, n, q$ this adapted mixing metric is

$$\operatorname{mix}(c, n, q) = \max_{\substack{0 \le \alpha < N, \\ 0 \le \beta < d_{\text{emb}}, \\ d_{\text{emb}} \le \gamma < 2 d_{\text{emb}}}} \left| \frac{\partial^2 \left( \boldsymbol{h}_c^{(2)} \right)_\alpha}{\partial \left( \boldsymbol{x}_q \right)_\beta \partial \left( \boldsymbol{x}_n \right)_\gamma} \right|,$$

which we plot in Figure 7. We see that gLSTM consistently exhibits greater maximum Hessian values than GCN, and that this collapses for GCN models above 8 neighbors, consistent with the drop in performance. As with the Jacobian ratios, we see plateauing and slow decrease of maximum Hessian values above the memory dimension, but these trends are less pronounced.

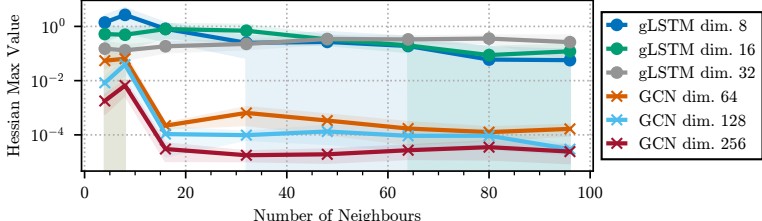

Figure 7: Mean of the maximum Hessian values for different gLSTM and GCN models, averaged across test set examples and different neighbor nodes. Standard deviation shaded.

## 5.3 LONG RANGE BENCHMARKS

Table 1: Mean and standard deviation of $\log_{10}(\text{MSE})$, averaged over 4 random weight initializations on the GPP tasks from Gravina et al. (2023), from which we report baselines. See Appendix D.1 for discussion of baseline choice. **Top** score in bold, second underlined. Lower is better.

| Method | Diam. | Ecc. | SSSP |
|---|---|---|---|
| GCN | 0.742 ± 0.047 | 0.846 ± 0.003 | 0.950 ± 0.000 |
| GAT | 0.822 ± 0.075 | 0.791 ± 0.022 | 0.695 ± 0.150 |
| GraphSAGE | 0.865 ± 0.40 | 0.286 ± 0.184 | 0.786 ± 0.021 |
| GIN | 0.613 ± 0.099 | 0.950 ± 0.001 | -0.541 ± 0.419 |
| GCNII | 0.529 ± 0.057 | 0.764 ± 0.036 | -1.132 ± 0.013 |
| DGC | 0.603 ± 0.005 | 0.826 ± 0.003 | -0.148 ± 0.023 |
| GRAND | 0.672 ± 0.049 | 0.660 ± 0.139 | -0.094 ± 0.340 |
| A-DGN | -0.546 ± 0.033 | 0.305 ± 0.118 | **-3.402 ± 0.137** |
| gLSTM (ours) | **-0.715 ± 0.030** | **-4.036 ± 0.311** | -2.836 ± 0.178 |
| - K-hop | 0.042 ± 0.123 | 0.673 ± 0.021 | -3.377 ± 0.142 |

We evaluate gLSTM on the Graph Property Prediction (GPP) tasks from Gravina et al. (2023) and the Long Range Graph Benchmark (LRGB) from Dwivedi et al. (2022). These benchmarks are both designed to require long range interactions to solve, and thus are an interesting test of the ability of gLSTM to overcome over-squashing and over-smoothing in real world tasks in order to facilitate long range interactions. Performance is reported in Table 1 and Table 2 respectively.

gLSTM achieves comfortably state of the art results on the Diameter and Eccentricity GPP tasks, and very strong performance on SSSP; notably SSSP is the only tested task in which k-hop decreases performance. LRGB results show that gLSTM achieves strong performance in Peptides-Func but relatively weak performance on Peptides-Struct. We hypothesize that the weaker performance on Peptides-Struct may be due to long-range interactions being less relevant for this task, which is very effectively solved by a few-layer GCN. See Appendix D.2 for gLSTM ablations on these benchmarks and Appendix D.5 for details around hyperparameters used.

Table 2: Mean and standard deviation on LRGB (Dwivedi et al., 2022), averaged over four random weight initializations. Baselines from the LRGB reevaluation of Tönshoff et al. (2024), K-hop methods from Arroyo et al. (2025), rewiring baseline from Barbero et al. (2024b). All methods adhere to a 500k parameter limit. **Top** score in bold, second underlined.

| Method | Peptides-Func | Peptides-Struct |
|---|---|---|
| | AP (↑) | MAE (↓) |
| GCN | 0.6860±0.0050 | **0.2460±0.0007** |
| GatedGCN | 0.6765±0.0047 | 0.2477±0.0009 |
| GINE | 0.6621±0.0067 | 0.2473±0.0017 |
| GPS | 0.6534±0.0091 | 0.2509±0.0014 |
| *K-hop methods* | | |
| kGCN-SSM | 0.6902±0.0022 | 0.2581±0.0003 |
| DRew-GCN | 0.6804±0.0144 | 0.2766±0.0019 |
| *Rewiring* | | |
| LASER | 0.6440±0.0010 | 0.3043±0.0019 |
| gLSTM (ours) | **0.7250±0.0023** | 0.2527±0.0015 |
| - K-hop | 0.6030±0.0096 | 0.2638±0.0010 |

## 6  CONCLUSION

In this work, we revisit over-squashing, disambiguating two bottleneck-related issues of sensitivity over-squashing and capacity over-squashing. We introduce a synthetic task that measures capacity over-squashing in isolation and we show that associative memory can improve MPNN capacity. The resulting architecture achieves strong results on real-world benchmarks.

**Future Work**    Many avenues remain open. Whereas the sensitivity issue of over-squashing has a mathematical basis via the node Jacobian, to our knowledge, the capacity issue does not. Theoretically quantifying this capacity could afford similar directions to those explored via sensitivity, establishing links to topology and model properties. With regards to architecture, we translate to a graph setting the gating and associative memory of xLSTM but do not retain the efficiency and parallel training, leaving open future work on more efficient MPNNs: we highlight that the recent work of Pöppel et al. (2025) achieves these efficiency gains in the specific case of directed acyclic graphs. Another potential avenue would be to apply our findings to prevention of issues of over-mixing and representational collapse (Barbero et al., 2024a; 2025; Arroyo et al., 2026) in Transformer architectures.

## REPRODUCIBILITY STATEMENT

We make available all of our code and experiment configurations to aid reproduction of results. Our experiments utilize the widely-used PyTorch Geometric GraphGym (You et al., 2020) framework which defines a standard framework for MPNN research.

For easiest reproduction of our results, please consult the `readme` in the code repository provided in Appendix D. The repository includes all necessary information to run the experiments: in particular, configs containing the hyperparameters used (also reported in Appendix D.5) and code for all plots used in the paper.

### ACKNOWLEDGMENTS

HB acknowledges funding support from the EPSRC Centre for Doctoral Training in Autonomous Intelligent Machines and Systems No. EP/S024050/1. MB is partially supported by the EPSRC Turing AI World-Leading Research Fellowship No. EP/X040062/1 and EPSRC AI Hub No. EP/Y028872/1.

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

# A    RELATIONSHIP TO COMPUTATIONAL (AND TOPOLOGICAL) BOTTLENECKS

Arnaiz-Rodriguez & Errica (2025) identify a separation in the over-squashing literature between the initial work of Alon & Yahav (2021) and the later work of Topping et al. (2022); Di Giovanni et al. (2023a). They point out that the original over-squashing definition of (Alon & Yahav, 2021) was associated with the computational tree, and Topping et al. (2022) later connected over-squashing to the existence of topological bottlenecks. They suggest that these definitions constitute different problems, and that – as a community – we should discard the term "over-squashing" and separate it into (at least) two separate terms: (1) computational tree bottlenecks and (2) topological bottlenecks (in the underlying graph).

This makes a valuable point: topological bottlenecks may or may not be involved when structural issues exist with the computational tree. The authors fairly point out that the umbrella term "over-squashing" has sometimes hidden some of the complexity of the problem, and the field may benefit from work identifying explicitly whether it is dealing with topological bottlenecks (as is the case with e.g. rewiring) or bottlenecks of the computational tree (e.g. adaptive message passing). We're unsure that framing these as *separate* is helpful, since topological bottlenecks must be mediated through the computational tree in order to impact message passing, but the point stands that some methods to combat over-squashing specifically target topological bottlenecks (and as such any changes to the computational graph are implicit), and some methods target general computational bottlenecks, and that this is not always clear.

Despite the complexity, we believe the term "over-squashing" still has utility as an umbrella term that describes issues that arise from the presence of bottlenecks and depth in the computational tree. The distinction made by Arnaiz-Rodriguez & Errica (2025) clarifies that sometimes these issues arise due to topological bottlenecks in the underlying graph, and sometimes they do not. The relevance of our work is then in exploring *how* this structure manifests as performance issues, and the main body of our paper argues that this is due to separate issues of **capacity** and **sensitivity**.

Precisely *how* issues of capacity and sensitivity relate to the structure of the computational tree and the presence of bottlenecks is complex, although we take initial steps to clarify this in our paper. It would **not** be correct to suggest for example that capacity issues correspond to computational bottlenecks and low sensitivity to topological bottlenecks. In particular:

- Our RingTransfer discussion of Section 3.1 highlights that computational bottlenecks can exist without capacity issues.

- Arnaiz-Rodriguez & Errica (2025) highlight that low sensitivity does not necessarily imply the existence of computational tree bottlenecks (or topological bottlenecks).

- On the other hand, both topological bottlenecks and general computational bottlenecks can cause low sensitivity. This is clear since the Equation (3) defines an upper bound on sensitivity in terms of the computational tree (defined by successive powers of the message-passing matrix), and we refer the reader to the work of Topping et al. (2022) for proofs in the case of specifically topological bottlenecks.

Additionally, while the discussion in this Appendix has so far focused on the computational tree, we highlight that the causes of the failure modes discussed in our work extend beyond structural issues: issues of capacity also depend at least on (1) information content of the task and (2) storage capacity of the model, as explored in Sections 3.1 and 3.2 respectively, and issues of sensitivity are highly dependent on the model dynamics, as explored in e.g. Arroyo et al. (2025); Heilig et al. (2025); Gravina et al. (2023; 2025).

# B    xLSTM UPDATE EQUATIONS

Beck et al. (2024) initially designed xLSTM as a combination of sLSTM blocks and mLSTM blocks - using scalar memory and matrix (associative) memory respectively. However, their follow up work (Beck et al., 2025) uses only mLSTM blocks, and these form the inspiration for gLSTM. Therefore, we will exclusively introduce the mLSTM block update equations in this section.

The update equations, presented in a similar manner to Section 4, are given below.

**State (and normalization) updates:**

$$\boldsymbol{C}^t = f^t \boldsymbol{C}^{t-1} + i^t \boldsymbol{v}^t \otimes \boldsymbol{k}^t \tag{4}$$

$$\boldsymbol{n}^t = f^t \boldsymbol{n}^{t-1} + i^t \boldsymbol{k}^t \tag{5}$$

$$m^t = \max\left(\tilde{f}^t + m^{t-1}, \tilde{i}^t\right) \tag{6}$$

**Query / Key / Value computation:**

$$\boldsymbol{q}^t = \boldsymbol{W}_q \boldsymbol{x}^t + \boldsymbol{b}_q \tag{7}$$

$$\boldsymbol{k}^t = \frac{1}{\sqrt{d}} \boldsymbol{W}_k \boldsymbol{x}^t + \boldsymbol{b}_k \tag{8}$$

$$\boldsymbol{v}^t = \boldsymbol{W}_v \boldsymbol{x}^t + \boldsymbol{b}_v \tag{9}$$

**Gate computation:**

$$i^t = \exp\left(\tilde{i}^t - m^t\right) \qquad\qquad \tilde{i}^t = \boldsymbol{w}_i^T \boldsymbol{x}^t + b_i \tag{10}$$

$$f^t = \exp\left(\tilde{f}^t + m^{t-1} - m^t\right) \qquad\qquad \tilde{f}^t = \boldsymbol{w}_f^T \boldsymbol{x}^t + b_f \tag{11}$$

$$\boldsymbol{o}^t = \sigma\left(\tilde{\boldsymbol{o}}^t\right) \qquad\qquad \tilde{\boldsymbol{o}}^t = \boldsymbol{W}_o \boldsymbol{x}^t + \boldsymbol{b}_o \tag{12}$$

**Output:**

$$\tilde{\boldsymbol{h}}^t = \boldsymbol{C}^t \boldsymbol{q}^t / \max\left\{\left|\boldsymbol{n}^{t\,T} \boldsymbol{q}^t\right|, 1\right\} \tag{13}$$

$$\boldsymbol{h}^t = \boldsymbol{o}^t \odot \tilde{\boldsymbol{h}}^t \tag{14}$$

## C   DATASET DETAILS

In table Table 3 we present a summary of the statistics of the datasets used throughout this work. We use the standard splits provided for each of the datasets, using the standard loading functionality from GraphGym (and PyTorch Geometric). For GPP (without a default implementation in PyTorch Geometric) we use the original splits of Gravina et al. (2023): 5120 graphs as training set, 640 as validation set, and 1280 as test set.

For the NAR task (for all neighbor values), we use a split of 8,000 train graphs, 1,000 validation graphs and 1,000 test graphs. We note that when testing gLSTM on this dataset, we vary the memory (matrix) dimension as described in the main text, and set the vector dimension to twice that of the matrix dimension.

## D   ADDITIONAL EXPERIMENTS

Our code for reproducing all experimental results in the main paper and appendices is publicly available at `https://github.com/HughBlayney/gLSTM`.

### D.1   ADDITIONAL GPP BASELINES

We note that due to a subtle PyTorch issue in the original GPP code implementation, normalization is not applied to the dataset targets. A refactor appears to have unknowingly fixed this issue in later iterations of the code so later experiments are run on a normalized variant of the dataset. Unfortunately, this results in unfair comparison, as results can be substantially different between the two variants of the dataset.

Therefore, in the main body of the paper we test only on the baselines provided in the original GPP paper (Gravina et al., 2023), as we are confident these use the un-normalized variant of the dataset, and this provides us with the largest number of baselines to test against. We additionally ensure

Table 3: Summary of datasets used throughout this work. Node and edge values are the average per graph, where applicable. LRGB statistics retrieved from Dwivedi et al. (2022), OGB statistics from Hu et al. (2020), Heterophilous from Platonov et al. (2023). GPP statistics computed; dataset from Gravina et al. (2023); Corso et al. (2020).

| Dataset | Group | Task | Setting | #Graphs | Nodes | Edges |
|---|---|---|---|---|---|---|
| Peptides-Func | LRGB | Graph cls. | Inductive | 15,535 | 150.94 | 307.30 |
| Peptides-Struct | LRGB | Graph reg. | Inductive | 15,535 | 150.94 | 307.30 |
| Diameter | GPP | Graph reg. | Inductive | 7,040 | 28.82 | 98.95 |
| Eccentricity | GPP | Node reg. | Inductive | 7,040 | 28.82 | 98.95 |
| SSSP | GPP | Node reg. | Inductive | 7,040 | 28.82 | 98.95 |
| arXiv | OGB | Node cls. | Transductive | 1 | 169,343 | 1,166,243 |
| Products | OGB | Node cls. | Transductive | 1 | 2,449,029 | 61,859,140 |
| Amazon-Ratings | Heterophilous | Node cls. | Transductive | 1 | 24,492 | 93,050 |
| Roman-Empire | Heterophilous | Node cls. | Transductive | 1 | 22,662 | 32,927 |
| Minesweeper | Heterophilous | Node cls. | Transductive | 1 | 10,000 | 39,402 |

that our method uses the same, un-normalized GPP variant. We separately test on the normalized version of the dataset, with results presented in Table 4 and hyperparameters in Table 14. This allows us to additionally test against a range of more recent methods that specifically target sensitivity over-squashing. We achieve state of the art results on the normalized variant of this benchmark.

Table 4: Mean and standard deviation of $\log_{10}(\text{MSE})$, averaged over 4 random weight initializations on the Normalized version of the GPP tasks from Gravina et al. (2023). Baselines reported from Arroyo et al. (2025); Eliasof et al. (2025). **Top** score in bold. Lower is better.

| Method | Diam. | Ecc. | SSSP |
|---|---|---|---|
| *Differential Equation Inspired GNNs* | | | |
| SWAN | $-0.598 \pm 0.115$ | $-0.074 \pm 0.219$ | $-3.543 \pm 0.083$ |
| PH-DGN | $-0.547 \pm 0.107$ | $-0.935 \pm 0.210$ | $-4.299 \pm 0.072$ |
| *K-hop methods* | | | |
| DRew-GCN | $-2.402 \pm 0.110$ | $-2.029 \pm 0.024$ | $-1.602 \pm 0.008$ |
| kGCN-SSM | $-3.075 \pm 0.055$ | $-4.265 \pm 0.178$ | $-3.604 \pm 0.029$ |
| GPS | $-0.512 \pm 0.043$ | $0.608 \pm 0.028$ | $-3.599 \pm 0.195$ |
| GRAMA | $-0.866 \pm 0.051$ | $-1.301 \pm 0.126$ | $-3.935 \pm 0.070$ |
| gLSTM (ours) | **-3.684 ± 0.057** | **-5.912 ± 1.116** | **-5.321 ± 0.603** |

## D.2 ABLATIONS

To identify what elements of the gLSTM architecture are most important for performance on these benchmarks, we perform ablations on the GPP and LRGB datasets. For LRGB, a task with a parameter limit, we ablate in two different settings: the first is simply removing the ablated component, for which results are presented in Table 6. The second is to scale the hidden dimension $h$ to keep the parameter count as close as possible to the 500k limit - i.e. when removing gating, this will correspondingly increase the hidden dimension. We include these experiments as they more accurately represent the reality of testing a model variant on a task with a parameter limit; these results are presented in Table 7. These two ablation settings show very similar results. GPP ablations are presented in Table 5.

We see that ablating gating only significantly reduces performance on Peptides-Func - other than this, it either leaves performance the same or in some cases, improves performance (GPP Ecc. in particular).

Table 5: Ablation of gLSTM performance on Diam, Ecc, and SSSP from the GPP benchmark. Mean and standard deviation are reported, averaged over four random weight initializations. Other than ablation, all other model settings are held constant; thus ablations with gating removed have reduced parameter count.

| Model | Diam. | Ecc. | SSSP |
|---|---|---|---|
| gLSTM | $-0.715 \pm 0.030$ | $-4.036 \pm 0.311$ | $-2.836 \pm 0.178$ |
| - Output gate | $-0.70 \pm 0.05$ | $-3.71 \pm 0.16$ | $-2.77 \pm 0.19$ |
| - Input gate | $-0.75 \pm 0.01$ | $-4.72 \pm 0.36$ | $-3.27 \pm 0.16$ |
| - Forget gate | $-0.71 \pm 0.03$ | $-4.30 \pm 0.21$ | $-3.14 \pm 0.07$ |
| - All gates | $-0.75 \pm 0.03$ | $-4.14 \pm 0.42$ | $-3.16 \pm 0.15$ |
| - K-hop aggregation | $0.04 \pm 0.12$ | $0.67 \pm 0.02$ | $-3.38 \pm 0.14$ |

Table 6: Ablation of gLSTM performance on Peptides-Func and Peptides-Struct from the LRGB. Mean and standard deviation are reported, averaged over four random weight initializations. Other than ablation, all other model settings are held constant; thus ablations with gating removed have reduced parameter count.

| Model | Peptides-Func | Peptides-Struct |
|---|---|---|
| | AP ($\uparrow$) | MAE ($\downarrow$) |
| gLSTM | $0.7250 \pm 0.0023$ | $0.2527 \pm 0.0015$ |
| - Output gate | $0.7086 \pm 0.0049$ | $0.2540 \pm 0.0016$ |
| - Input gate | $0.7186 \pm 0.0029$ | $0.2524 \pm 0.0027$ |
| - Forget gate | $0.7236 \pm 0.0063$ | $0.2522 \pm 0.0011$ |
| - All gates | $0.7180 \pm 0.0088$ | $0.2526 \pm 0.0012$ |
| - Positional encoding | $0.7208 \pm 0.0072$ | $0.2539 \pm 0.0036$ |
| - K-hop aggregation | $0.6030 \pm 0.0096$ | $0.2638 \pm 0.0010$ |

Table 7: Ablation of gLSTM performance on Peptides-Func and Peptides-Struct from the LRGB. Mean and standard deviation are reported, averaged over four random weight initializations. All methods adhere to a 500k parameter limit such that hidden dimension varies to keep parameter count as close to this as possible.

| Model | Peptides-Func | Peptides-Struct |
|---|---|---|
| | AP ($\uparrow$) | MAE ($\downarrow$) |
| gLSTM | $0.7250 \pm 0.0023$ | $0.2527 \pm 0.0015$ |
| - Output gate | $0.7202 \pm 0.0056$ | $0.2537 \pm 0.0011$ |
| - Input gate | $0.7193 \pm 0.0110$ | $0.2518 \pm 0.0027$ |
| - Forget gate | $0.7148 \pm 0.0107$ | $0.2545 \pm 0.0043$ |
| - All gates | $0.7188 \pm 0.0060$ | $0.2528 \pm 0.0035$ |
| - Positional encoding | $0.7211 \pm 0.0062$ | $0.2601 \pm 0.0017$ |
| - K-hop aggregation | $0.6030 \pm 0.0096$ | $0.2638 \pm 0.0010$ |

Table 8: Mean and standard deviation on Heterophilic Datasets, averaged over four random weight initializations.

| Model | Roman-empire | Amazon-ratings | Minesweeper |
|---|---|---|---|
| | Acc ↑ | Acc ↑ | AUC ↑ |
| **MPNNs** | | | |
| GAT | $80.87_{\pm 0.30}$ | $49.09_{\pm 0.63}$ | $92.01_{\pm 0.68}$ |
| GAT (LapPE) | $84.80_{\pm 0.46}$ | $44.90_{\pm 0.73}$ | $93.50_{\pm 0.54}$ |
| GAT (RWSE) | $86.62_{\pm 0.53}$ | $48.58_{\pm 0.41}$ | $92.53_{\pm 0.65}$ |
| Gated-GCN | $74.46_{\pm 0.54}$ | $43.00_{\pm 0.32}$ | $87.54_{\pm 1.22}$ |
| GCN | $73.69_{\pm 0.74}$ | $48.70_{\pm 0.63}$ | $89.75_{\pm 0.52}$ |
| GCN (LapPE) | $83.37_{\pm 0.55}$ | $44.35_{\pm 0.36}$ | $94.26_{\pm 0.49}$ |
| GCN (RWSE) | $84.84_{\pm 0.55}$ | $46.40_{\pm 0.55}$ | $93.84_{\pm 0.48}$ |
| CO-GNN$(\Sigma, \Sigma)$ | $91.57_{\pm 0.32}$ | $51.28_{\pm 0.56}$ | $95.09_{\pm 1.18}$ |
| CO-GNN$(\mu, \mu)$ | $91.37_{\pm 0.35}$ | $54.17_{\pm 0.37}$ | $97.31_{\pm 0.41}$ |
| SAGE | $85.74_{\pm 0.67}$ | $53.63_{\pm 0.39}$ | $93.51_{\pm 0.57}$ |
| **Graph Transformers** | | | |
| Exphormer | $89.03_{\pm 0.37}$ | $53.51_{\pm 0.46}$ | $90.74_{\pm 0.53}$ |
| NAGphormer | $74.34_{\pm 0.77}$ | $51.26_{\pm 0.72}$ | $84.19_{\pm 0.66}$ |
| GOAT | $71.59_{\pm 1.25}$ | $44.61_{\pm 0.50}$ | $81.09_{\pm 1.02}$ |
| GPS | $82.00_{\pm 0.61}$ | $53.10_{\pm 0.42}$ | $90.63_{\pm 0.67}$ |
| GPS$_{\text{GCN+Performer}}$ (LapPE) | $83.96_{\pm 0.53}$ | $48.20_{\pm 0.67}$ | $93.85_{\pm 0.41}$ |
| GPS$_{\text{GCN+Performer}}$ (RWSE) | $84.72_{\pm 0.65}$ | $48.08_{\pm 0.85}$ | $92.88_{\pm 0.50}$ |
| GPS$_{\text{GCN+Transformer}}$ (LapPE) | OOM | OOM | $91.82_{\pm 0.41}$ |
| GPS$_{\text{GCN+Transformer}}$ (RWSE) | OOM | OOM | $91.17_{\pm 0.51}$ |
| GT | $86.51_{\pm 0.73}$ | $51.17_{\pm 0.66}$ | $91.85_{\pm 0.76}$ |
| GT-sep | $87.32_{\pm 0.39}$ | $52.18_{\pm 0.80}$ | $92.29_{\pm 0.47}$ |
| Polynormer | $92.55_{\pm 0.30}$ | $54.81_{\pm 0.49}$ | $97.46_{\pm 0.36}$ |
| **Heterophily-Designated GNNs** | | | |
| CPGNN | $63.96_{\pm 0.62}$ | $39.79_{\pm 0.77}$ | $52.03_{\pm 5.46}$ |
| FAGCN | $65.22_{\pm 0.56}$ | $44.12_{\pm 0.30}$ | $88.17_{\pm 0.73}$ |
| FSGNN | $79.92_{\pm 0.56}$ | $52.74_{\pm 0.83}$ | $90.08_{\pm 0.70}$ |
| GBK-GNN | $74.57_{\pm 0.47}$ | $45.98_{\pm 0.71}$ | $90.85_{\pm 0.58}$ |
| GloGNN | $59.63_{\pm 0.69}$ | $36.89_{\pm 0.14}$ | $51.08_{\pm 1.23}$ |
| GPR-GNN | $64.85_{\pm 0.27}$ | $44.88_{\pm 0.34}$ | $86.24_{\pm 0.61}$ |
| H2GCN | $60.11_{\pm 0.52}$ | $36.47_{\pm 0.23}$ | $89.71_{\pm 0.31}$ |
| JacobiConv | $71.14_{\pm 0.42}$ | $43.55_{\pm 0.48}$ | $89.66_{\pm 0.40}$ |
| **Graph SSMs** | | | |
| GMN | $87.69_{\pm 0.50}$ | $54.07_{\pm 0.31}$ | $91.01_{\pm 0.23}$ |
| GPS + Mamba | $83.10_{\pm 0.28}$ | $45.13_{\pm 0.97}$ | $89.93_{\pm 0.54}$ |
| GRAMA$_{\text{GCN}}$ | $88.61_{\pm 0.43}$ | $53.48_{\pm 0.62}$ | $95.27_{\pm 0.71}$ |
| MP-SSM | $90.91_{\pm 0.48}$ | $53.65_{\pm 0.71}$ | $95.33_{\pm 0.72}$ |
| **Ours** | | | |
| gLSTM (ours) | $88.12 \pm 0.35$ | $51.98 \pm 0.45$ | $92.08 \pm 0.88$ |
| + K-hop | $83.48 \pm 0.30$ | $52.25 \pm 0.31$ | $87.55 \pm 0.78$ |

## D.3 Heterophilic Benchmarks

We additionally evaluate gLSTM on the heterophilic datasets of Platonov et al. (2023), shown in Table 8. Hyperparameters and sweeps are documented in Appendix D.5 (Table 13).

gLSTM achieves good MPNN-level performance, but is outperformed by some Graph Transformer architectures. Notably, the addition of K-hop aggregation here actually *harms* performance significantly in two of the three datasets. We suggest this may be due in part to the lack of long-range dependencies in these datasets, whereas the GPP and LRGB datasets evaluated in the main text are designed specifically to contain long range dependencies. This explanation would imply that K-hop provides a useful inductive bias for long range dependencies specifically.

Table 9: Mean and standard deviation on OGBN, averaged over four random weight initializations.

| Method | Arxiv | Products |
|---|---|---|
| GCN | $71.74 \pm 0.29$ | $75.64 \pm 0.21$ |
| ChebNet | $73.27 \pm 0.23$ | - |
| ChebNetII | $72.32 \pm 0.23$ | - |
| GraphSAGE | $71.49 \pm 0.27$ | $78.29 \pm 0.16$ |
| GAT | $72.02 \pm 0.44$ | $79.45 \pm 0.59$ |
| NodeFormer | $59.90 \pm 0.42$ | $72.93 \pm 0.13$ |
| GraphGPS | $70.97 \pm 0.41$ | OOM |
| GOAT | $72.41 \pm 0.40$ | $82.00 \pm 0.43$ |
| NAGphormer | $70.13 \pm 0.55$ | $73.55 \pm 0.21$ |
| Exphormer | $72.44 \pm 0.28$ | OOM |
| SGFormer | $72.63 \pm 0.13$ | $74.16 \pm 0.31$ |
| Polynormer | $73.46 \pm 0.16$ | $83.82 \pm 0.11$ |
| gLSTM (ours) | $71.91 \pm 0.32$ | $81.52 \pm 0.29$ |

### D.4 LARGE GRAPH BENCHMARKS AND SCALING EXPERIMENTS

This section investigates the scaling of gLSTM to large graphs, for which we use the Arxiv and Products node classification datasets from the Open Graph Benchmark (Hu et al., 2020). `ogbn-arxiv` comprises 169,343 nodes and 1,166,243 edges, while `ogbn-products` comprises 2,449,029 nodes and 61,859,140 edges. We evaluate gLSTM on these graphs without K-hop aggregation, which would be prohibitively expensive to compute on these graphs. However, as noted in Appendix D.3, it appears that K-hop aggregation is of particular importance for long-range dependencies, so it is unclear whether it would improve performance on these benchmarks anyway.

Results on these datasets are presented in Table 9. Hyperparameters and sweeps are documented in Appendix D.5 (Table 12).

We highlight that although the model which achieves best validation-set performance on `ogbn-arxiv` is large, thus incurring high runtime and VRAM usage, significantly smaller models achieve comparable results with only a small decrease in accuracy: in particular, a model with memory dimension 8, 8 heads and 4 layers achieves an accuracy of $71.31 \pm 0.25$.

We use `ogbn-arxiv` to empirically measure gLSTM epoch time and VRAM scaling with the number of heads and memory dimension: these results are visualized in Figure 8, reporting both absolute values and multiples of the corresponding metric for a GCN model with 4 layers and hidden dimension 256 on the same dataset. As expected, these values appear to scale linearly with the number of heads and quadratically with the (matrix) memory dimension.

We highlight that we have not optimized any of the gLSTM code for CUDA efficiency, so the relative epoch times and VRAM usages may be overstated, and future improvements are likely possible.

### D.5 HYPERPARAMETERS

In Tables 10 and 11 we present the hyperparameter sweeps and chosen hyperparameters for results in the main body of the paper: GPP and LRGB respectively. In Tables 12 to 14 we present the hyperparameter sweeps and chosen hyperparameters for the normalized version of GPP (Appendix D.1), Heterophilic benchmarks (Appendix D.3) and OGBN benchmarks (Appendix D.4) respectively.

### D.6 OVERSMOOTHING AND LONG RANGE DEPENDENCIES

We test empirically that gLSTM is able to learn long range dependencies by evaluating on the RingTransfer task introduced in Di Giovanni et al. (2023a). Results for gLSTM, GCN and GNN-SSM (Arroyo et al., 2025) are shown for various ring sizes (and corresponding number of message passing layers) in Figure 9.

Table 10: Hyperparameter sweeps for gLSTM on LRGB tasks. In bold are the hyperparameters that achieved the best validation set performance, and thus were those used in the main results of the paper. Note that hidden dimension was not directly swept over, as this was maximized for each configuration such that the model remained within the 500k parameter budget. Due to compute limitations, hyperparameter sweeps were not exhaustive, but used *Weights and Biases* Bayesian Optimization routine with Hyperband early termination.

| Hyperparameter | Peptides-Func | Peptides-Struct |
|---|---|---|
| Memory Dimension | 8, 16, **32** | 8, **16**, 32 |
| Number of Heads | 1-**2**-8 | 1-**5**-8 |
| Message Passing Layers | 10-**27**-50 | 4-**23**-40 |
| Input Norm Type | **Layer** | **Layer**, None |
| Hidden Norm Type | **Group** | **Group** |
| Act. Func. (between block) | GeLU, ReLU, **None** | GeLU, **ReLU**, None |
| Dropout | **0.1** | **0.0**, 0.1, 0.2 |
| Hidden Dimension | **45** | **42** |

Table 11: Hyperparameter sweeps for gLSTM on GPP tasks. In bold are the hyperparameters that achieved the best validation set performance, and thus were those used in the main results of the paper. Hyperparameters were tested exhaustively via grid search.

| Hyperparameter | Diam. | Ecc. | SSSP |
|---|---|---|---|
| Memory Dimension | 8, **16** | **8**, 16 | 8, **16** |
| Number of Heads | **1**, 2, 3, 4 | 1, 2, 3, **4** | **1**, 2, 3, 4 |
| Message Passing Layers | 1, 5, 10, **20** | 1, 5, **10**, 20 | 1, 5, **10**, 20 |
| Input Norm Type | **None** | **None** | **None** |
| Hidden Norm Type | **Group** | **Group** | **Group** |
| Act. Func. (between block) | Tanh, **ReLU**, None | Tanh, **ReLU**, None | **Tanh**, ReLU, None |
| Dropout | **0.0** | **0.0** | **0.0** |
| Hidden Dimension | **10**, 20, 30 | 10, **20**, 30 | **10**, 20, 30 |

Table 12: Hyperparameter sweeps for gLSTM on OGBN (Hu et al., 2020) tasks. Bold entries indicate the hyperparameters that achieved the best validation performance and were used in the main experiments.

| Hyperparameter | ogbn-arxiv | ogbn-products |
|---|---|---|
| Memory Dimension | 8,16,**32** | **8**,16,32 |
| Number of Heads | **4**,8 | 4,**8** |
| Message Passing Layers | **4**,6,8,10 | 4,**6**,8,10 |
| Input Norm Type | **Batch** | **Layer** |
| Hidden Norm Type | **Group** | **Group** |
| Act. Func. (between block) | **ReLU** | **ReLU** |
| Dropout | **0.5** | **0.5** |
| Hidden Dimension | **256** | **256** |

Table 13: Hyperparameter sweeps for gLSTM on the heterophilous datasets of Platonov et al. (2023). Bold entries indicate the hyperparameters selected based on validation performance. Sweep ranges were identical across datasets.

| Hyperparameter | Amazon Ratings | Minesweeper | Roman Empire |
|---|---|---|---|
| Memory Dimension | 8,16,**32** | **8**,16,32 | 8,16,**32** |
| Number of Heads | 4,**8** | **4**,8 | 4,**8** |
| Message Passing Layers | **4**,6,8,10,12 | 4,6,8,**10**,12 | **4**,6,8,10,12 |
| Input Norm Type | **Batch** | **Batch** | **Batch** |
| Hidden Norm Type | **Group** | **Group** | **Group** |
| Act. Func. (between block) | **ReLU** | **ReLU** | **ReLU** |
| Dropout | **0.5** | **0.2** | **0.5** |
| Hidden Dimension | **512** | **64** | **512** |

Table 14: Hyperparameter sweeps for gLSTM on the Normalized-GPP tasks: sweep is performed only over Normalized-Diam and these parameters are used for all other experiments. In bold are the hyperparameters that achieved the best validation set performance, and thus were those used for test set evaluation.

| Hyperparameter | Normalized-Diam. |
|---|---|
| Memory Dimension | 8, **16**, 32 |
| Number of Heads | 1-**4**-8 |
| Message Passing Layers | 4-**10**-40 |
| Input Norm Type | **Layer** |
| Hidden Norm Type | **Group** |
| Act. Func. (between block) | GeLU, ReLU, **None** |
| Dropout | 0.0 **0.1** 0.2 |
| Hidden Dimension | **512** |

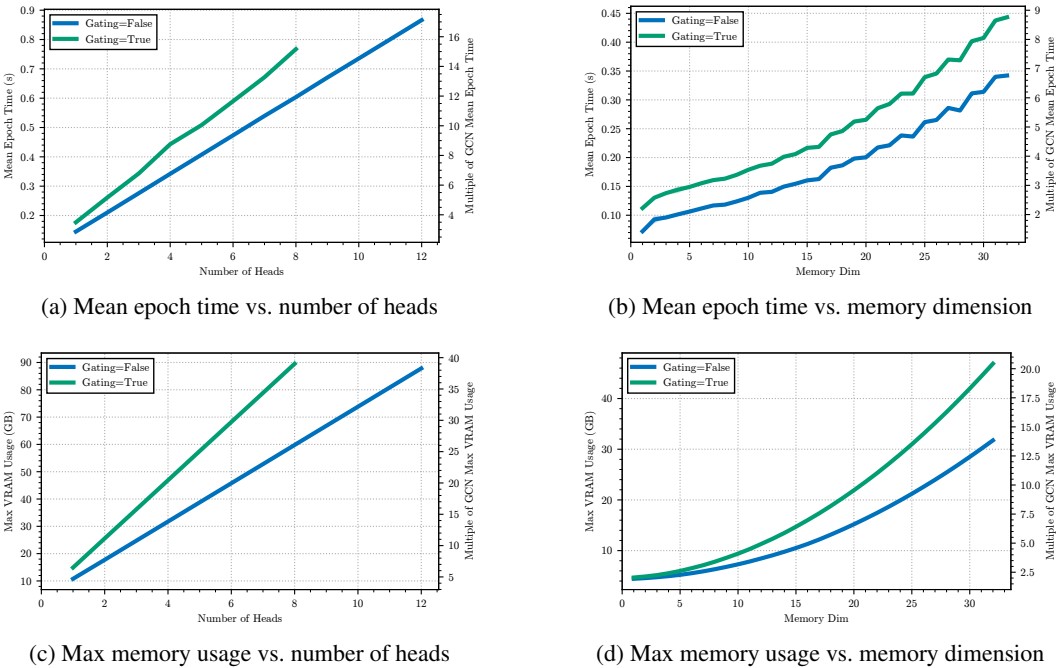

(a) Mean epoch time vs. number of heads

(b) Mean epoch time vs. memory dimension

(c) Max memory usage vs. number of heads

(d) Max memory usage vs. memory dimension

Figure 8: Scaling behavior of gLSTM, empirically reported on the `ogbn-arxiv` dataset (Hu et al., 2020). Mean epoch time and max memory usage is reported, both in absolute terms and relative to GCN with 4 layers and hidden dimension of 256. All parameters other than that being varied are fixed to that of the top-performing model: see Table 12.

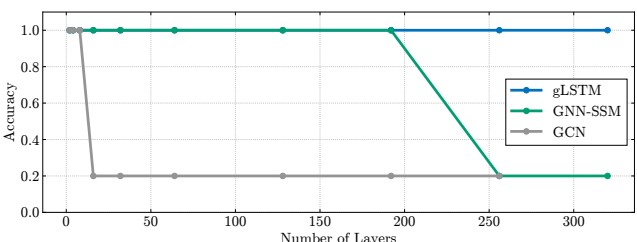

Figure 9: Performance on the RingTransfer task.

## D.7 ADDITIONAL NAR CLASSIFICATION RESULTS

In this section, we present various additional results from the NAR task presented in the main body of the paper.

We visualize the Jacobian norms - separated by selected vs background nodes - for the mixed aggregation strategies used in the main paper in Figure 10. This is, in effect, the more granular plot of Figure 6b.

### D.7.1 COMPARISON AGAINST (GRAPH) TRANSFORMER BASELINES

In addition to the MPNN comparisons presented in the main body of the paper and Appendix D.7.3, we compare gLSTM against two Transformer baselines: GraphGPS (Rampášek et al., 2022) and a regular Transformer (Vaswani et al., 2017) block utilizing softmax attention. These results are visualized in Figure 11.

From this figure, we see that softmax attention with a sufficiently high hidden dimension is able to solve the task perfectly, as expected. Curiously, GraphGPS fails significantly earlier, at 32 neighbors for all hidden dimensions tested: while we unsure of the precise reason for this, we speculate that

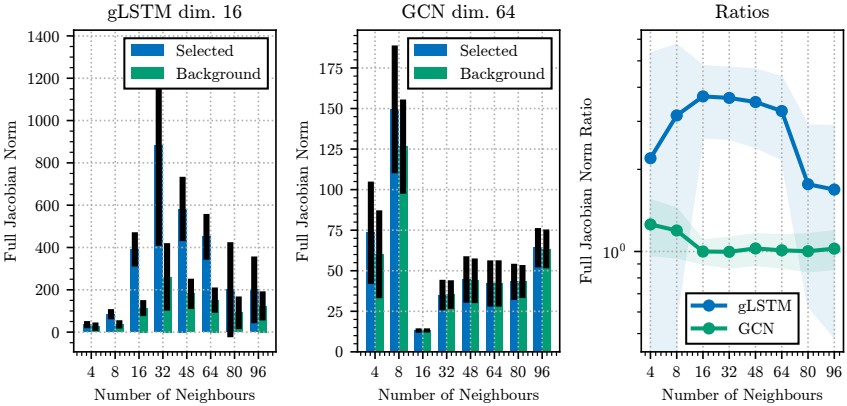

Figure 10: **Left**: Mean Jacobian norms for gLSTM of memory dimension 16, with varying number of neighbors in the NAR task, separated by whether the neighbor node corresponds to the given query (selected) or not (background). **Middle**: Same, with GCN of hidden dimension 64. **Right**: Mean ratios of Jacobian norms for selected nodes to background nodes, for these two models. Standard deviation visualized in bar chart error bars and line chart shaded area.

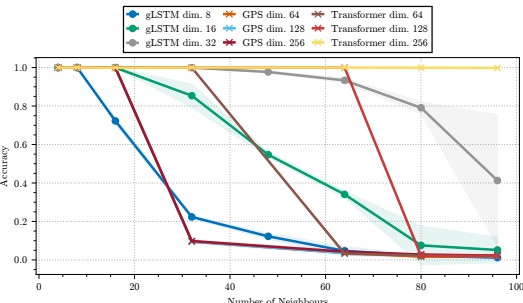

Figure 11: Comparison of gLSTM performance against Transformer baselines: GPS (Rampášek et al., 2022) and a standard Transformer baseline with softmax attention. Both of these use Laplacian positional encodings appended to the node feature vectors. Test-set mean accuracy is shown, with standard deviation shaded, averaged over 3 runs. Note that gLSTM uses K-hop aggregation here, whereas GPS does not.

the GCN aspect of the architecture may be actively harming performance. We note however that GraphGPS does outperform all MPNNs tested, other than gLSTM.

It is also interesting to note that at lower dimensions, softmax attention reaches a neighbor count at which it also starts to fail. It seems clear that this arises simply from the initial embedding of the integer key, query and value symbols: when the size of these alphabets exceeds half the hidden dimension (recall half of the embedding vector is used to encode the key, the other half the query), it is no longer possible for each symbol to have a learned orthogonal vector and there will be overlap.

### D.7.2 TRAINING CURVES: FAILURE AT NAR IS AN ISSUE OF GENERALIZATION

This section investigates further what exactly causes GCN and other models to fail at NAR. We observe in Figure 12 that for several models, above the point where test set accuracy decreases, train set accuracy remains high: thus, the failure mode seems to be one of *generalization*. We suggest that this is consistent with the capacity intuition: at the point at which the model is no longer able to separably store the neighbor representations, it collapses instead to memorizing the aggregated neighbor representation and corresponding value. Since these specific combinations are not shared with the test set, this does not generalize.

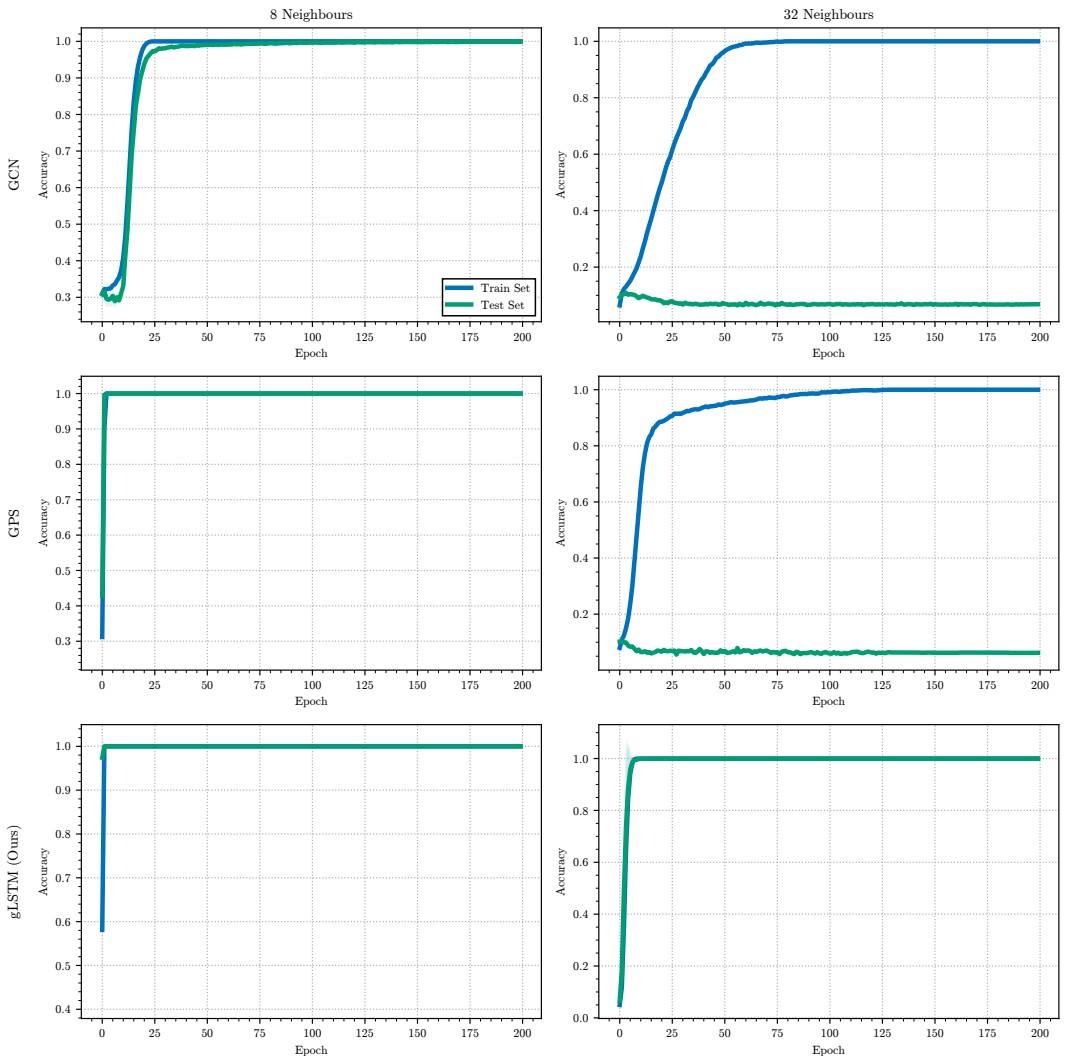

Figure 12: Train and test set accuracies plotted against epoch count. Left column contains the results for experiments using 8 neighbor nodes, right column for 32 neighbors. Top row visualises GCN results, second GPS, third gLSTM. gLSTM is the only one that remains able to generalize at 32 neighbors (and beyond).

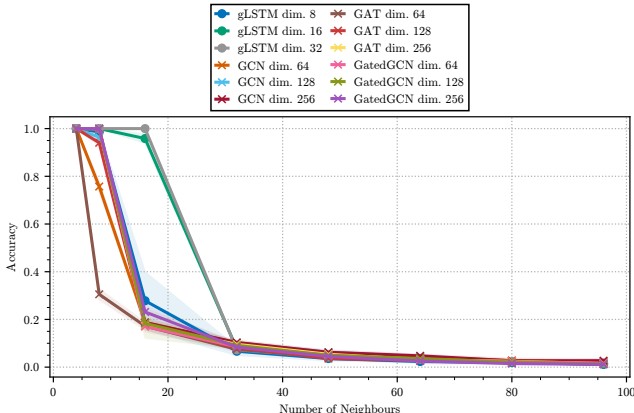

Figure 13: NAR Accuracy where all models do *not* use K-hop aggregation, for an expanded set of models.

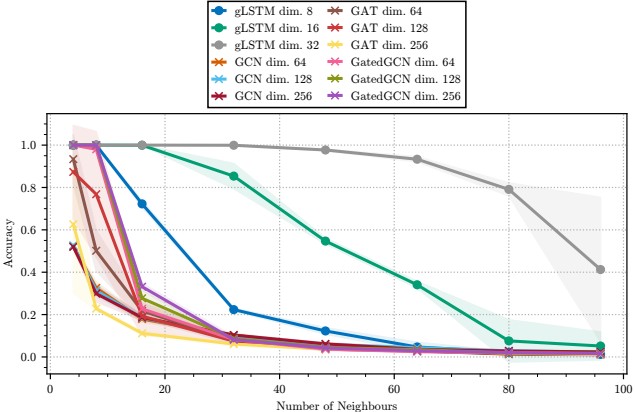

Figure 14: NAR Accuracy where all models *do* use K-hop aggregation, for an expanded set of models.

### D.7.3 PERFORMANCE SEPARATED BY AGGREGATION STRATEGY

We next separate out no-K-hop and K-hop aggregation, and plot results for a larger set of models, in Figures 13 and 14 respectively.

We additionally verify that the number of layers is not the reason behind GCN being unable to solve NAR at higher neighbor counts. Figure 16 visualizes the performance of GCN models with hidden

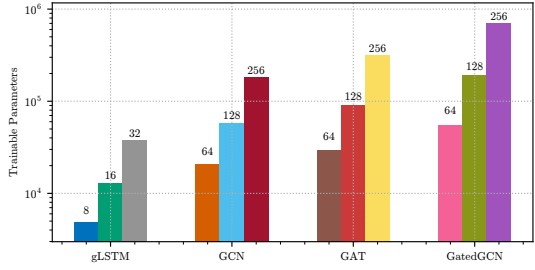

Figure 15: Number of trainable parameters for the expanded set of models tested in Figures 13 and 14.

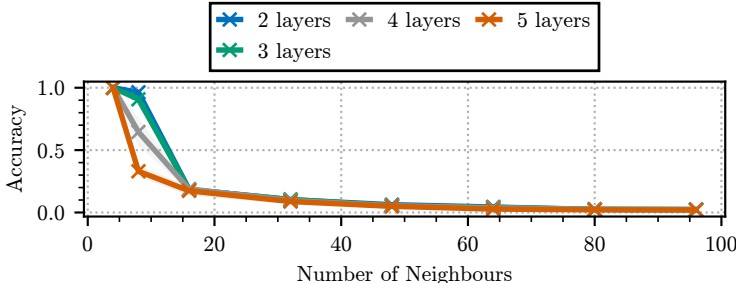

Figure 16: NAR Accuracy for GCN of hidden dimension 128, no K-hop, for varying numbers of GCN layers.

dimension 128 and various layer counts; it transpires that 2 layers performs best out of the tested layer counts.

### D.7.4 ADDITIONAL SENSITIVITY METRIC RESULTS

We plot in this section the sensitivity metric trends of gLSTM vs GCN, both using K-hop aggregation.

Figure 17a visualizes the Jacobian norms for different model sizes and numbers of neighbors; Figure 17b shows the ratios between selected and background node Jacobian norms. Figure 18 separates out the Jacobian norms for gLSTM memory dimension 16 and GCN hidden dimension 64. Figure 19 visualizes the Hessian mixing metric for all models.

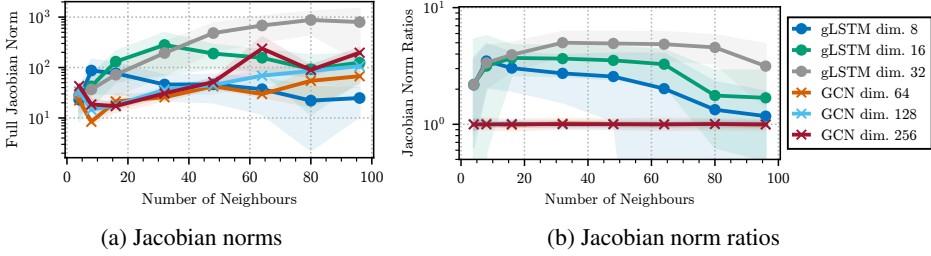

(a) Jacobian norms                    (b) Jacobian norm ratios

Figure 17: Left: Average Jacobian norms for different gLSTM and GCN models, with varying number of neighbors in the NAR task. Right: The ratio between the Jacobian norms of the selected (key corresponds to query) to background (key is different from query) neighbor nodes, for the different models - see Figure 18. This plot differs from that in the main body of the paper in that both gLSTM and GCN use K-hop aggregation.

### D.8 NEIGHBOR ASSOCIATIVE RECALL REGRESSION RESULTS

In this section, we present results for the *regression* variant of the NAR task presented in the main body of the paper. We refer to this as Neighbor Associative Recall Regression (NARR).

Similarly to NAR, for a given neighborhood size $N$ we create a graph of $N + 3$ nodes. This graph consists of $N$ "neighbor" nodes, a central node to which they are all connected, and an intermediate node connected to the central node and a "query" node connected only to the intermediate node.

Each of the neighbor nodes has a feature vector representing a key and a value. The values consist of a fixed-dimensional vector of length $V$ where each element is randomly sampled from a standard normal distribution. The keys are each unique one-hot vectors of dimension $N$. The query node's feature vector contains a single one-hot vector, equal to one of the one-hot vectors of the neighbor nodes. The target of the graph is for the central node to predict the *value* of the neighbor node, corresponding to the key that matches the query node. Each node $u$ is therefore equipped with an input feature vector $x \in \mathbb{R}^{V+2N}$, where the first $V$ elements comprise the value, the next $N$ elements the key and the final $N$ elements the query. Where a node does not have one of these features, the

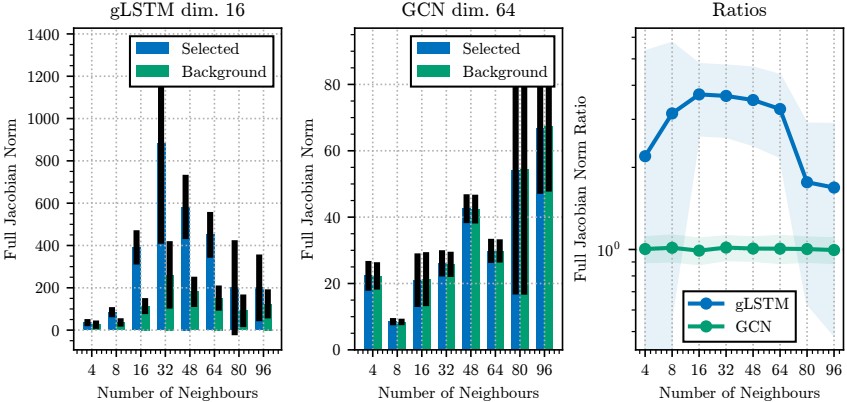

Figure 18: Left: Jacobian norms for gLSTM of memory dimension 16, with varying number of neighbors in the NAR task, separated by whether the neighbor node corresponds to the given query (selected) or not (background). Middle: Same, with GCN of hidden dimension 64. Right: Ratios of Jacobian norms for selected nodes to background nodes, for these two models. This plot differs from that in the main body of the paper in that both gLSTM and GCN use K-hop aggregation.

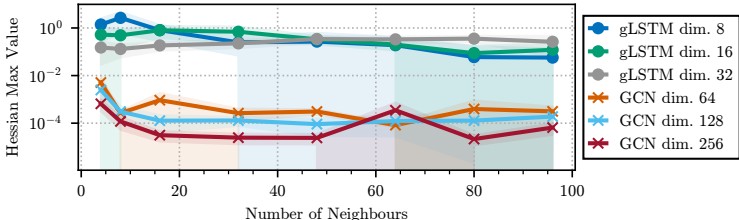

Figure 19: Mean of the maximum Hessian values for different gLSTM and GCN models, averaged across test set examples and different neighbor nodes. This plot differs from that in the main body of the paper in that both gLSTM and GCN use K-hop aggregation.

vector elements are set to zero. We note the use of one-hot encoding for keys and values means that the first linear layer of the model acts as a learned embedding function, where multiplication with the one-hot encoding simply selects the corresponding column of the weight matrix. For our experiments, we use $V = 16$.

Since the value vectors lack the sparsity of NAR, this appears to be a "harder" task in the sense that it is more taxing on memory capacity. This means that some of the over-squashing trends are more defined, particularly trends in sensitivity-based measures - see Appendix D.8.1. However, our experiments suggest that the regression target means that NARR becomes too hard for vector-memory MPNNs to effectively solve, visible in Figures 20 and 21.

Performance (MSE) for NARR is shown in Figures 20 and 21 for no-K-hop and K-hop aggregation respectively. We note that the performance curves in Figure 21 look similar to those obtained by the sequence modeling variant of this experiment in Schlag et al. (2021). The number of trainable parameters is shown in Figure 22.

### D.8.1 RELATIONSHIP TO OVER-SQUASHING SENSITIVITY METRICS

As with NAR in the main paper, we visualize the behavior of sensitivity-based over-squashing metrics for different neighbor counts and different models. Similarly to the main paper, we compare gLSTM using K-hop aggregation and GCN without. We note that – perhaps due to the increased difficulty of the task – the trends discussed in Section 5.2 are actually *more* pronounced for the NARR task.

Figure 23a visualizes the Jacobian norms for different model sizes and numbers of neighbors; Figure 23b shows the ratios between selected and background node Jacobian norms. Figure 24 separates

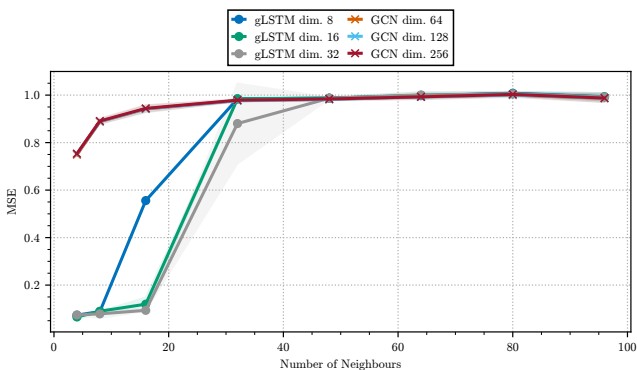

Figure 20: NARR MSE where all models do *not* use K-hop aggregation.

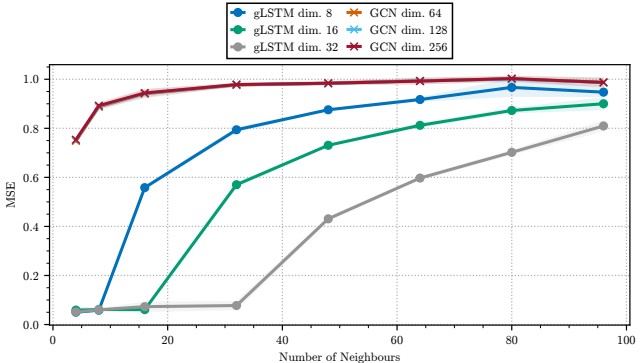

Figure 21: NAR Accuracy where all models *do* use K-hop aggregation.

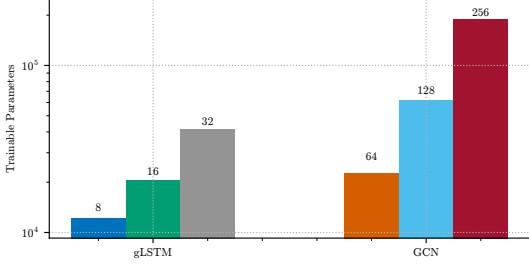

Figure 22: Number of trainable parameters for the expanded set of models tested in Figures 20 and 21.

out the Jacobian norms for gLSTM memory dimension 16 and GCN hidden dimension 64. Figure 25 visualizes the Hessian mixing metric for all models.

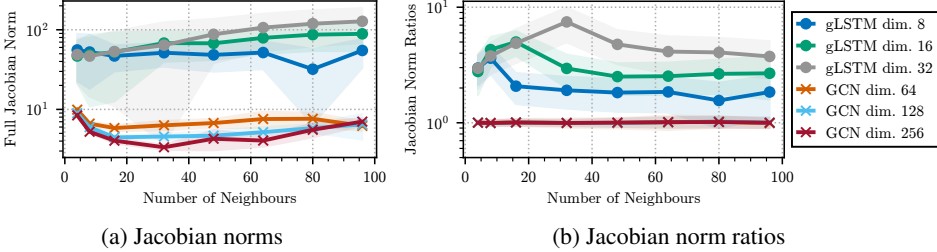

(a) Jacobian norms          (b) Jacobian norm ratios

Figure 23: Left: Average Jacobian norms for different gLSTM and GCN models, with varying number of neighbors in the NARR task. Right: The ratio between the Jacobian norms of the selected (key corresponds to query) to background (key is different from query) neighbor nodes, for the different models - see Figure 24.

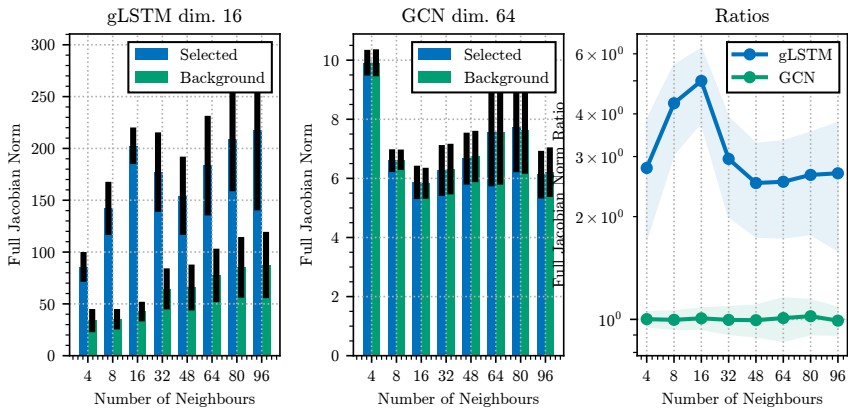

Figure 24: Left: Jacobian norms for gLSTM of memory dimension 16, with varying number of neighbors in the NARR task, separated by whether the neighbor node corresponds to the given query (selected) or not (background). Middle: Same, with GCN of hidden dimension 64. Right: Ratios of Jacobian norms for selected nodes to background nodes, for these two models.

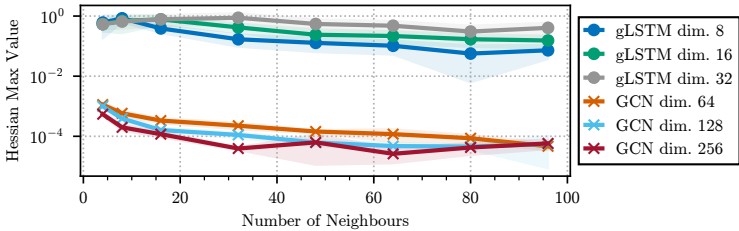

Figure 25: Mean of the maximum Hessian values for different gLSTM and GCN models, averaged across test set examples and different neighbor nodes.

We note that the sensitivity difference between selected and background nodes is particularly stark here, even more so for classification-based NAR; gLSTM consistently shows a sharp drop-off in Figure 23b at the memory dimension, and GCN maintains a ratio remarkably close to unity. This closely aligns with the performance of these models, Figure 21 demonstrates that gLSTM performance begins to drop off quickly when the number of neighbors matches the memory dimension, and Figure 20 demonstrates that GCN is never able to solve the task, for any tested number of neighbors.

We hypothesize that the strong impact of the K-hop aggregation on the model's ability to selectively recall - particularly visible for NARR - may partially explain the dramatic performance decrease

when ablating this aggregation, discussed in Appendix D.2. We note that, while gLSTM consistently demonstrates superior performance to GCN, the improved performance is most striking when additionally using K-hop aggregation; it appears that the inductive bias introduced by the K-hop aggregation is particularly suited to the selective recall required by this task.

