# OpenReview forum: "gLSTM: Mitigating Over-Squashing by Increasing Storage Capacity"
_ICLR.cc/2026/Conference — ICLR 2026 Poster_

### Official Review · Reviewer_6qip · 2025-10-29

**Soundness:** 3
**Presentation:** 3
**Contribution:** 3
**Rating:** 6
**Confidence:** 2

**Summary:**

This paper redefines *over-squashing* in Graph Neural Networks (GNNs) as two distinct failure modes: **insufficient sensitivity** and **storage capacity saturation**, arguing that existing evaluations often conflate the two. The authors introduce a synthetic task called **Neighbor Associative Recall (NAR)** that isolates and measures capacity alone. Building on this task, they propose a message-passing architecture called **gLSTM**, which incorporates **matrix-based associative memory** and **xLSTM-style gating**. Additionally, they employ **K-hop aggregation** to enhance long-range interactions. The proposed method demonstrates clear capacity advantages on the NAR task and achieves competitive or state-of-the-art results on several GPP and LRGB benchmarks. The authors also analyze the relationship between capacity and sensitivity using Jacobian and Hessian-based measures.

**Strengths:**

- The paper clearly separates “capacity” and “sensitivity” when analyzing over-squashing, and introduces a shallow, controllable synthetic task (NAR) that isolates and measures storage capacity. The experiments and quantitative analyses strongly support this formulation.


- The proposed gLSTM architecture combines matrix-based associative memory with fast-weight (outer-product) updates, which is well aligned with the message-passing paradigm in graphs; the derivation and implementation details are sound and well explained.


- The method shows strong performance on multiple long-range dependency benchmarks, with comprehensive ablation studies (on gating, K-hop aggregation, positional encoding, etc.) and full hyperparameter tables.


- The paper is reproducible and transparent: code and configurations are released, and the training framework is standardized and well documented.

**Weaknesses:**

- Coupling between K-hop aggregation and capacity claims:

In real-world benchmarks, the performance gains may stem not only from increased storage capacity but also from enhanced sensitivity due to the denser computation graph created by K-hop aggregation. This coupling makes it difficult to attribute improvements solely to capacity. The paper should include controlled comparisons that fix K-hop settings while varying only the memory and gating components, alongside sensitivity metrics.

- Lack of efficiency and scalability analysis:

The matrix memory in gLSTM increases computational and memory costs quadratically with the hidden dimension. Although parameter counts are aligned across models, the paper does not report training/inference time or peak memory usage. Adding throughput and GPU memory comparisons with standard baselines (GCN, GIN, GCNII, GPS, etc.) would strengthen the empirical claims.

**Questions:**

- Under the same K-hop and parameter budget, how does gLSTM compare to equivalent non-memory MPNNs (wider or deeper) in both performance and sensitivity metrics such as Jacobian or mixed second-order derivatives?

- Please report training/inference time and peak memory consumption, and provide scaling curves showing how efficiency varies with memory dimension and the number of heads.

---

> ### Author Response · Authors · 2025-11-21
>
> Thank you for taking the time to read our work and provide this thoughtful feedback. We have organised our response below according to the weaknesses raised in your review.
>
> ## Training / Inference Time and Peak Memory Consumption
> This is a very good comment that was missing in our previous draft of the paper. We have now included this in our updated draft and we agree this makes the analysis significantly more complete: we have tested significantly larger graphs (ogbn-arxiv and ogbn-products in particular), demonstrating that gLSTM (without k-hop) remains performant at this scale: please see Appendix D4, Table 9. We have also plotted, according to your suggestion, scaling curves for epoch runtime and maximum VRAM usage, both in absolute terms and compared to a GCN model: these can be found in Figure 8.
>
> ## Comparison to equivalent non-memory MPNNs in performance and sensitivity metrics
> This information is documented in the Appendix, as it requires a large number of plots (and therefore space) to communicate. We draw your attention in particular to
> 1. Figures 13 and 14, which compares a wide range of MPNNs using the same aggregation strategy (non k-hop and k-hop respectively)
> 2. Figures 17 and 18, which compare sensitivity metrics when all models are using K-hop aggregation.
>
> As NAR is designed with 2-hop MPNNs in mind we do not comprehensively compare against variable-depth MPNNs, but we also highlight Figure 16, which plots GCN performance for a range of depths: this demonstrates that increased depth does not help with NAR performance.
>
> In response to
> > the performance gains may stem not only from increased storage capacity but also from enhanced sensitivity due to the denser computation graph created by K-hop aggregation.
>
> We note that gLSTM with k-hop achieves significantly improved results when compared to other k-hop methods, such as DRew and kGNN-SSM - see Table 2 and (added in our updated draft) Table 4. These models provide a K-hop “baseline”, suggesting that the additional performance improvement is due to associative memory.
>
> If there is any analysis still missing that you would like to see, please let us know.

---

> > ### Comment · Reviewer_6qip · 2025-11-24
> >
> > Thanks for the author's response. I maintain my positive score.

---

### Official Review · Reviewer_QVwX · 2025-10-31

**Soundness:** 3
**Presentation:** 2
**Contribution:** 2
**Rating:** 2
**Confidence:** 4

**Summary:**

The paper proposes gLSTM, a graph neural architecture that integrates an LSTM-like gating and memory mechanism into the message-passing process of GNNs. The authors argue that by maintaining global and local memory states, the model can alleviate over-squashing problem.

**Strengths:**

1. The idea of combining LSTM-style gating with graph message passing is conceptually intuitive and aligns with the goal of controlling information flow.

2. The experiments on the synthetic Neighbor Associative Recall (NAR) task provide some evidence that the model can store multiple neighbor features effectively.

3. The overall writing is easy to follow.

**Weaknesses:**

1. The main novelty of the paper is the adaptation of xLSTM to GNNs. While the motivation is reasonable, most design components are directly inherited from prior LSTM or xLSTM works. The paper feels more like an engineering extension rather than a fundamentally new GNN architecture. The authors should clearly articulate what is truly new in the design beyond re-using LSTM elements.

2. Although the idea of using LSTM to control message passing is intuitive, the technical motivation for several design choices is insufficiently explained. For instance, the roles of 𝑛𝑢 and 𝑚𝑢 on graphs are not clearly defined—are they merely borrowed normalization and stabilization terms from xLSTM, or do they have specific graph-theoretic meaning? This ambiguity makes it difficult to understand how these terms contribute to mitigating over-squashing.

3. The paper lacks a clear description of the datasets and settings. Although LRGB and synthetic graphs are mentioned, there is no systematic list of datasets, splits used or how baselines are chose. It remains unclear how results generalize to standard benchmarks such as OGB datasets or larger real-world graphs.

4. The main analysis of over-squashing is performed on the synthetic NAR task. While this toy setting helps isolate capacity effects, it is unclear whether the claimed improvements hold on realistic graphs. Results on real-world datasets are necessary to substantiate the claim.

5. Recent Graph Transformers and other scalable GNNs have been shown to alleviate over-squashing via global attention mechanisms. However, these models are not compared, leaving it unclear how gLSTM performs relative to current state-of-the-art methods.

6. The proposed design may introduce significant computational overhead due to additional gating, normalization, and memory matrices. No runtime or complexity analysis is provided, especially for large graphs such as papers100M.

7. Although the empirical intuition is sound, the paper lacks a rigorous theoretical justification for why the LSTM-style mechanism fundamentally reduces over-squashing. A formal analysis would strengthen the claims.

8. The experiments only include graph-level classification and synthetic tasks. There are no results on node classification or link prediction, which are precisely the domains where over-squashing is more prominent. This omission limits the paper’s empirical impact.

**Questions:**

Please refer to the weakness.

---

> ### Author Response · Authors · 2025-11-21
>
> Thank you for taking the time to read our paper, we greatly appreciate this feedback and the opportunity to improve our work.
>
> ## Lack of Novelty
> As you note, part of our paper involves adapting the xLSTM architecture to a graph setting. However, this is not intended as the main novelty of the work: we intend instead for the primary novelty of our work to be recharacterising over-squashing as a combination of separate issues of sensitivity and capacity. As part of this, we introduce a novel synthetic task to measure this capacity in isolation from sensitivity issues, and we also introduce an example architecture which we show increases model capacity, and performs better at this synthetic task. As such, we intend the gLSTM architecture to be more of a proof of concept, demonstrating that capacity is a model feature which can be improved through techniques such as associative memory.
>
> We note however that there are, in our view, interesting and novel aspects of the gLSTM architecture. The coupled nature of the vector and matrix node memories - vector state at the previous layer updates the matrix state, which in turn is queried to obtain the next vector state - is, to our knowledge, novel and empirically performs well. This means that gLSTM is not a simple adaptation of xLSTM [1]: no graph structure that can be designed such that the two are equivalent. Similarly, this separates our work from other sequence-to-graph adaptations such as GRED [2], which do not use this additional recurrence.
>
> ## Lack of Real World Benchmark Validation
> You mention
> > It remains unclear how results generalize to standard benchmarks such as OGB datasets or larger real-world graphs.
>
> While we evaluated on LRGB tasks (predicting properties of molecular graphs) we acknowledge that the real-world validation of our work was not sufficiently comprehensive. Therefore, we have evaluated on multiple new benchmarks, which we include in the updated draft.
>
> We add evaluations on
> - Large-scale node classification tasks: ogbn-products and ogbn-arxiv. See Appendix D4 and Table 9. Products comprises a graph with approximately 2.5M nodes, demonstrating that gLSTM scales to large graphs.
> - Heterophilic benchmarks: amazon-ratings, minesweeper and roman-empire. See Appendix D3 and Table 8.
>
> We also include empirical runtime and memory usage analysis on ogbn-arxiv, reporting both absolute values and a comparison to GCN - see Appendix D4 and Figure 8. We hope that this goes some way to address several of your concerns: 1) lack of real world evaluation 2) lack of node classification evaluation and 3) lack of runtime and complexity analysis on large graphs.
>
> ## Lack of Comparison to Graph Transformers
> We have included new results on the NAR task, showing the performance of GraphGPS [3] and a pure transformer architecture: see Figure 11 in Appendix D.7.1. We discover as expected that a pure transformer - for a sufficiently high hidden dimension - is able to solve the task perfectly, as the attention mechanism lends itself particularly well to the key-value recall task. However, this review prompted us to discover an interesting result, which is that GPS does not exhibit the same behaviour: although it performs better than MPNNs, its performance deteriorates far earlier than pure softmax attention, and earlier even than gLSTM (16 and 32 dimensions). We hope that these comparisons provide a representative sample of the “scalable GNNs… [with] global attention mechanisms” that you reference in your review, but if you would like to see evaluation against any other models in particular, please let us know.
>
> We additionally include a variety of graph transformer baselines (including GraphGPS[3], Exphormer[4], NAGphormer[5], Polynormer[6]) on our real-world evaluations: see Tables 8 and 9.
>
> ## Other
> We acknowledge your feedback that our work lacks a clear description of datasets and settings. We use standard splits, but we will make this clear and collate all of this information in one table - along with dataset properties - in a future draft.
>
> We agree that there is still work to be done on a rigorous theoretical justification on the capacity issue of over-squashing, and we similarly noted this in the conclusion of the paper. We leave this for future work however, and we note that we are unaware of rigorous theoretical analysis of the corresponding issues in the sequence modeling literature.
>
> ## References
> [1] Beck, Maximilian, et al. "xlstm: Extended long short-term memory."
>
> [2] Ding, Yuhui, et al. "Recurrent distance filtering for graph representation learning."
>
> [3] Rampášek, Ladislav, et al. "Recipe for a general, powerful, scalable graph transformer."
>
> [4] Shirzad, Hamed, et al. "Exphormer: Sparse transformers for graphs."
>
> [5] Chen, Jinsong, et al. "NAGphormer: A tokenized graph transformer for node classification in large graphs."
>
> [6] Deng, Chenhui, Zichao Yue, and Zhiru Zhang. "Polynormer: Polynomial-expressive graph transformer in linear time."

---

> > ### Author Response · Authors · 2025-12-02
> >
> > Just a brief note that we have now added a table and description of the datasets and settings in our most recent draft, as referenced in our previous comment. Please see Appendix C and Table 3.

---

### Official Review · Reviewer_ekHn · 2025-11-01

**Soundness:** 3
**Presentation:** 2
**Contribution:** 3
**Rating:** 6
**Confidence:** 3

**Summary:**

This paper revisits over-squashing in message-passing neural networks by emphasizing a **capacity** perspective in addition to the usual **sensitivity** view. It introduces a synthetic evaluation, **Neighbor Associative Recall (NAR)**, designed to isolate node storage capacity in a shallow setting. Building on sequence-modeling ideas, the authors propose **gLSTM**, a GNN architecture with associative (matrix) memory and gating, paired with **K-hop aggregation**. Experiments on NAR and standard graph benchmarks suggest that increasing node-level storage capacity improves long-range reasoning while remaining competitive downstream.

**Strengths:**

1. **Clear conceptual separation.** The work carefully distinguishes capacity-related vs. sensitivity-related over-squashing and makes the capacity lens the central object of study.
2. **Purpose-built synthetic task.** NAR is a targeted probe for node storage limits that avoids confounding depth-driven sensitivity effects common in prior tests; the setup and metrics make the evaluation legible.
3. **Methodological innovation.** gLSTM adapts associative memory and exponential gates (in the spirit of xLSTM) to graphs, coupled with K-hop aggregation to broaden accessible context. The design choices are motivated and technically transparent.
4. **Empirical rigor and diagnostics.**
    * On NAR, gLSTM maintains **high recall** and then **gracefully saturates** near the memory limit, consistent with the capacity hypothesis.
    * On downstream benchmarks (e.g., GPP Diameter/Eccentricity; LRGB Peptides-Func), results are strong/competitive within a parameter budget.
    * Jacobian/Hessian-style sensitivity analyses help separate capacity effects from sensitivity/optimization effects.

**Weaknesses:**

1. **Limited Theoretical Guarantees**:
   As noted in the conclusion, the capacity notion is supported primarily through intuition and empirical performance. There is currently no theoretical quantification or formal bounding of node storage capacity for different architectures (gLSTM, GCN).

2. **Dependency on K-hop Aggregation**:
   Much of gLSTM's performance gain (Appendix B.2, Table 3 and onward) appears to come from K-hop aggregation, raising the question of whether capacity improvements are due to associative memory or simply enhanced information accessibility.
3. **Baseline Limitations**:
   The experimental section primarily compares gLSTM to GCN and occasionally to a few other standard models (Table 1); Particularly, the lack of evaluation against strong geometric or spectrum-preserving approaches is an empirical gap.

**Questions:**

1. How does gLSTM’s time and memory footprint scale as node and edge counts grow? In practice, do you observe memory bottlenecks that differ between sparse and dense graphs.

2. What motivated the choice of K-hop neighborhood specifically, and could alternative aggregation schemes provide similar or even better empirical benefits? Could the authors provide more detail/access to K-hop ablation results in the main body?

3. Could the authors give more interpretative diagnosis when gLSTM or GCN "breaks down" (e.g., NAR neighbor count exceeding memory)?

4. Please include strong capacity-oriented and rewiring baselines for both NAR and downstream tasks, or provide a clear and compelling justification for their omission. In the absence of such comparisons or reasoning, I will have to lower the score due to insufficient empirical validation.

---

> ### Author Response · Authors · 2025-11-21
>
> Thank you for taking the time to review our paper and leaving this detailed and thoughtful feedback. We appreciate your summary of the strengths of our paper - these closely align with our intentions with this work. And we also appreciate the chance to improve our work by addressing your comments.
>
> ## Dependency on k-hop
> Your comment and question around K-hop are both valid. It is true that much of gLSTM’s performance gain on the LRGB tasks and GPP appears to be linked to the k-hop aggregation.
>
> To help address this we have now added a large range of additional experiments that provide some nuance to this observation: we ablate k-hop on a range of heterophilic datasets (see Appendix D3, Table 8) and discover that in 2 of 3 cases, K-hop significantly harms performance. We also evaluate on the OGBN arxiv and products dataset and discover that even without k-hop, gLSTM achieves strong performance (see Appendix D4, Table 9). This leads us to suggest that K-hop is particularly valuable for LRGB and GPP **because of the existence of long-range dependencies.**
>
> In response to
>
> >Much of gLSTM's performance gain (Appendix D.2, Table 5 and onward) appears to come from K-hop aggregation, raising the question of whether capacity improvements are due to associative memory or simply enhanced information accessibility.
>
> we note that gLSTM with k-hop achieves significantly improved results when compared to other k-hop methods, such as DRew and kGNN-SSM - see Table 2 and (added in the new draft) Table 4. These models provide a K-hop “baseline”, suggesting that the additional performance improvement is due to associative memory.
>
> In response to
>
> > What motivated the choice of K-hop neighborhood specifically, and could alternative aggregation schemes provide similar or even better empirical benefits?
>
> 1) it was already shown to empirically perform well (DRew, GNN-SSM, ChebNet, to a certain extent GRED) and
> 2) as you noted, it affords “enhanced information accessibility” and a highly connected graph structure.
>
> We were then surprised by how effective this aggregation method transpired to be for the gLSTM architecture, but this is interesting in itself: in the paper we speculate that it may provide an inductive bias that works particularly well with associative memory. You ask whether alternative aggregation schemes could provide similar or even better empirical benefits: we suspect that the answer to this is yes, but this is not something that we investigated in this work. We hope that our k-hop analysis provides evidence that highly-connected aggregation strategies may empirically perform well (especially when paired with associative memories), and future work may take this further.
>
> You note that it would be good to have more access to K-hop ablations in the main body of the paper: for the moment we have moved the LRGB k-hop ablation to the main table (Table 2) in the paper (and this ablation is already included for GPP), but we would be happy to take suggestions for how this could be further improved.
>
> ## Scaling Behaviour
> Although the matrix memory of gLSTM adds memory overhead (addressed below), the scaling behaviour with graph size and density will be equivalent to other MPNNs as the underlying message passing mechanism is the same.
>
> We demonstrate that gLSTM is able to scale to larger graphs by including results on ogbn-arxiv and ogbn-products (Appendix D4, Table 9), the latter of which comprises approximately 2.5M nodes. We also include empirical runtime and memory usage analysis on ogbn-arxiv, reporting both absolute values and a comparison to GCN (Appendix D4, Figure 8).
>
> ## Extended Baselines
> You mention including capacity-oriented and rewiring baselines for both NAR and downstream tasks, or else to see reasons for their omission. To address this, we have included a rewiring baseline in Table 2 and additional results of graph transformers (which we view as a capacity-oriented method) and regular transformers on the NAR task, with interesting results - please see Appendix D.7.1 and Figure 11.
>
> We do not include rewiring methods (other than k-hop) on NAR: NAR is meant to test situations where the first message passing round requires the storage of a large amount of information (with no way to filter what information should be stored), and the second message passing round requires the model to selectively retrieve from this information (both k-hop and regular aggregation satisfy this). Arbitrary rewiring methods would not necessarily preserve this behaviour, making it unclear what we are measuring.
>
> If you have other capacity oriented or specific rewiring methods that you would like to see a comparison to, please let us know.
>
> ## Other
> We agree with your comment that there is no theoretical quantification of formal bounding of node storage capacity, but we leave this to future work. We note that we are unaware of rigorous theoretical analysis of the corresponding issues in the sequence modeling literature.

---

> > ### Comment · Reviewer_ekHn · 2025-11-25
> >
> > I thank the authors for the detailed rebuttal and the substantial changes in the revised draft. The new version directly addresses several of my main concerns.
> >
> > My main remaining reservation is that the notion of “capacity” is still framed empirically and architecturally rather than via a formal theory or explicit bounds. The authors acknowledge this as an open problem, and I agree that a full theoretical treatment is likely beyond the scope of a single paper; I now see this more as a limitation of the current theory landscape than a flaw specific to this work.
> >
> > Overall, the revised manuscript significantly strengthens the case for gLSTM and for the proposed capacity-oriented perspective on over-squashing. As a result, I will maintain my positive overall score.

---

### Official Review · Reviewer_6dpm · 2025-11-01

**Soundness:** 2
**Presentation:** 3
**Contribution:** 2
**Rating:** 4
**Confidence:** 4

**Summary:**

The paper identifies two distinct mechanisms behind “over-squashing” in MPNNs (message-passing graph neural networks): (i) sensitivity over-squashing and (ii) capacity over-squashing. It introduces a shallow synthetic task (Neighbor Associative Recall, NAR) that isolates capacity from sensitivity, and proposes gLSTM, an MPNN whose node states are augmented with an associative-memory matrix updated via fast-weight outer products. On NAR, gLSTM retains high recall until the number of neighbors equals the memory dimension, while vanilla GCN fails much earlier. gLSTM also obtains good performance on some real-world benchmarks.

**Strengths:**

1. The paper is well-written and well-motivated with the synthetic tasks. The problem targeted is of real significance.
2. It is wise to disentangling capacity from sensitivity to give an in-depth analysis of capacity over-squashing.

**Weaknesses:**

1. The proposed method is a simple combination of existing techniques, particularly applying xlstm to graph structures. The tech novelty should be further strengthened.
2. The proposed method may suffer from high computational overhead, there should be relevant analysis. Moreover, it is unclear the size of the tested benchmarks. It is advised to show that the proposed method can be applied to graphs with more than 1M nodes.

**Questions:**

How would the proposed method perform when applied to deep mpnns?

---

> ### Author Response · Authors · 2025-11-21
>
> Thank you for taking the time to read our work and providing us with this feedback. Your summary of our paper captures effectively what we had intended to communicate, so we appreciate that. We have organised our response below according to the weaknesses raised in your review.
>
> ## Technical Novelty
> While it is true that our proposed method adheres quite closely to some of the design choices of xLSTM, there are nontrivial novel aspects of the architecture that we chose based on empirical performance and the challenges of adapting to a graph structure. The coupled nature of the vector and matrix node memories - where the vector state at the previous layer is used to update the matrix state, which in turn is queried to obtain the next vector state - is, to our knowledge, novel and appears to empirically perform well. This means, for example, that gLSTM is not a simple adaptation of xLSTM[1], as there is no graph structure that can be designed such that the two are equivalent. Similarly, this separates our work from other sequence-to-graph adaptations such as GRED[2], which do not use this additional recurrence.
>
> However, we highlight (as you summarised) that the introduction of the gLSTM architecture is not intended as the main contribution of our paper: but rather we intend instead for the primary novelty of our work to be a recharacterisation of the over-squashing problem in GNNs as a combination of separate issues of sensitivity and capacity. As part of this, we introduce a novel synthetic task to measure this capacity in isolation from sensitivity issues, and we also introduce an example architecture which we show increases model capacity, and performs better at this synthetic task. In this way, we intend the gLSTM architecture to be more of a proof of concept - that capacity is a model feature which can be improved through techniques such as associative memory.
>
> ## Computational Overhead Analysis
> Your comment that gLSTM may suffer from high computational overhead is a good point, and one that required further investigation. In our updated draft of the paper, we demonstrate that our method scales to graphs with more than 1M nodes by including results on significantly larger graphs (ogbn-arxiv and ogbn-products, the latter of which comprises approximately 2.5M nodes) - see Appendix D4 and Table 9. We also conduct an empirical scaling analysis of epoch runtime and maximum memory usage as we modify the number of gLSTM heads and the memory dimension, reporting both absolute values and a comparison to GCN - see Figure 8. Alongside this we conduct several additional experiments on heterophilic graphs, details of which can be found in Appendix D3, Table 8.
>
> ## How Does the Proposed Method Perform when applied to deep MPNNs?
> We answer this question in two ways, but if we have misunderstood the purpose of your question, please let us know, and we will respond appropriately.
>
> gLSTM scales very effectively to a large number of layers, partly due to the use of block structure, normalisation layers and avoidance of over-squashing via increased memory capacity. This is evidenced by e.g. the strong performance we observe on peptides-func, in which gLSTM performs optimally with 27 layers.
>
> The NAR task is specifically designed for 2-layer GNNs, but we additionally test what happens when deeper GNNs are applied: Figure 16 demonstrates that GCN doesn’t perform any better when more layers are added.
>
> ## References
> [1] Beck, Maximilian, et al. "xlstm: Extended long short-term memory."
>
> [2] Ding, Yuhui, et al. "Recurrent distance filtering for graph representation learning."

---

> > ### Comment · Reviewer_6dpm · 2025-11-26
> >
> > Thanks for the reply. I've raised my rating.

---

### Official Review · Reviewer_mSY2 · 2025-11-04

**Soundness:** 3
**Presentation:** 3
**Contribution:** 3
**Rating:** 6
**Confidence:** 4

**Summary:**

This paper addresses the over-squashing phenomenon in GNNs by focusing specifically on storage capacity as distinct from sensitivity. The authors introduce a new synthetic benchmark, NAR, designed to isolate and directly measure capacity over-squashing. Inspired by xLSTM and fast weight programmers from sequence modeling, the authors propose a new GNN architecture (gLSTM) to boost storage capacity. Empirical evaluations demonstrate the effectiveness of gLSTM on the NAR task and several long-range graph benchmarks.

**Strengths:**

- The paper provides a rigorous re-examination of over-squashing, explicitly distinguishing between storage capacity and sensitivity, with careful argumentation. It clarifies and extends understanding within the community.

- The NAR task is a well-designed, controlled test for probing capacity bottlenecks. It enables discrimination between capacity and sensitivity effects, addressing limitations of prior synthetic benchmarks.

- The paper provides detailed mathematical descriptions of both baseline GNNs and the proposed gLSTM mechanism.

- The authors have conducted extensive ablations and sensitivity analyses to support their claims. Jacobian and Hessian analyses also enhance interpretability.

- The experiments show that gLSTM achieves state-of-the-art or competitive performance in both the synthetic NAR task and real-world benchmarks.

**Weaknesses:**

1. The related work section omits several directly relevant recent studies [1, 2].  Besides, the authors are encouraged to discuss the difference between over-smoothing [3, 4] and over-squashing.

2. While the authors propose the NAR task and empirically study capacity limits, there is a lack of formal and quantitative capacity analysis.

3. The move from vector to matrix memory increases parameter counts and computational complexity. There is little analysis or discussion about it.

4. Direct empirical comparison to a broader set of anti-over-squashing methods is missing.

Refs:

[1] Schreier-Coset Graph Propagation, Arxiv 2025.

[2] Over-Squashing in GNNs and Causal Inference of Rewiring Strategies, Arxiv 2025.

[3] SkipNode: On alleviating performance degradation for deep graph convolutional networks, TKDE 2024.

[4] Dropedge: Towards deep graph convolutional networks on node classification, ICLR 2019.

**Questions:**

Please address the weaknesses.

---

> ### Author Response · Authors · 2025-11-21
>
> Thank you for your time reading our work and providing this feedback. We appreciate your comments on the strength of our re-examination of over-squashing and how this helps improve the understanding of the community. We have organised our response below according to the weaknesses raised in your review.
>
> ## Scaling Behaviour
> We have included more experiments on significantly larger graphs, demonstrating that gLSTM scales to graphs of approximately 2.5M nodes (ogbn-products) and achieves strong performance: please see Appendix D4 and Table 9. We have also included additional experiments investigating the scaling of runtime and maximum memory usage with the number of heads and the memory dimension on another large graph dataset, ogbn-arxiv: see Figure 8. We hope that this helps to address the issue you have identified of a lack of analysis or discussion around the scaling of computational complexity and activated parameter count.
>
> ## More Over-Squashing Baselines
> We have included several more results and experiments to address this lack of comparison to anti-over-squashing methods:
> 1. We have included results of GraphGPS and regular softmax attention on the NAR task, which mitigate over-squashing via global attention: see Appendix D.7.1 and Figure 11
> 2. We have included new numbers on an updated version of the GPP task, which allows us to compare to a larger range of methods designed to combat sensitivity over-squashing: see Table 4.
> 3. We have included a rewiring baseline for LRGB, Table 2
>
> If there are any other specific anti-over-squashing methods you think that it is valuable for us to compare against, please let us know.
>
> ## Related Work
> Thank you for drawing our attention to these studies - with the time pressure of running additional experiments we have not yet had the time to integrate them but we will do so in a future updated draft.

---

> > ### Comment · Reviewer_mSY2 · 2025-11-26
> >
> > Thanks for the reply, and I will keep my positive rating.

---

### Author Response · Authors · 2025-11-21
**Summary of Changes in Response to Reviewers**

We thank the reviewers for their time reviewing our paper, and for all of the helpful feedback provided. There were many insightful questions and comments raised which we have worked hard to address in this updated draft of our paper: this process has been highly valuable, and we believe it has improved the clarity and robustness of our work, while surfacing several interesting new insights.

The revised draft that we have uploaded colors in blue the relevant changes that address reviewer comments, and we also reference specific sections of this draft in our individual replies to the reviewers. To summarise the relevant changes:

**Scaling investigations**:
All reviewers requested additional analysis on the scalability of this method. To address this we added evaluations of gLSTM on significantly larger graphs: ogbn-arxiv and ogbn-products, comprising 169,343 and 2,449,029 nodes respectively - see Appendix D4 and Table 9. We report epoch runtime and maximum VRAM usage on ogbn-arxiv for a range of gLSTM head counts and memory dimensions, thus investigating scaling behaviour with these hyperparameters. We report both absolute values and values relative to those of a comparable GCN model acting on the same graph. Please see Appendix D4 and Figure 8 for this analysis.

**Comparison on a larger range of real world benchmarks**:
As requested by reviewer QVwX, we evaluated gLSTM on a much wider range of real world benchmarks, ensuring that we also investigated node classification tasks. In addition to the large-scale OGBN datasets mentioned above, we tested on a range of common heterophilic datasets: Amazon Ratings, Roman Empire and Minesweeper. Results on these benchmarks are presented in Appendix D3, Table 8.

**Additional baselines to target over-squashing**:
Reviewers QVwX, ekHn and mSY2 request comparison to a larger set of baselines, particularly those that target over-squashing. We have taken several steps to address this:
- Additional comparison to graph transformer architectures on the NAR task: see Appendix D.7.1, Figure 11.
- Comparison to a range of MPNNs that specifically target sensitivity over-squashing in Appendix D1, Table 4
- Comparison to a wide range of graph transformer architectures in Tables 8 and 9
- Addition of a rewiring baseline in the LRGB evaluation of Table 2

Beyond this summary, we reference in our individual replies many smaller changes we have made to address specific feedback.

Thank you again to the reviewers for their suggestions. We hope that these revisions address the concerns raised, and will be taken into account in evaluating our work. Where concerns remain or have not been adequately addressed in our revision, we would welcome additional feedback so we can work to address that.

---

### Meta-Review · Area_Chair_vLRY · 2026-01-07

**Summary:**

This work re‑frames over‑squashing in message‑passing GNNs as two distinct failure modes: (a) sensitivity loss and (b) storage‑capacity saturation. By treating capacity as the primary bottleneck it introduces a synthetic benchmark, Neighbor Associative Recall (NAR), that cleanly isolates how much information a node can store and retrieve without depth‑induced sensitivity effects. To address capacity over‑squashing the authors propose gLSTM, a graph neural architecture that augments each node’s vector state with an associative‑memory matrix updated via fast‑weight outer‑product programming and gated by xLSTM‑style gates; K‑hop aggregation is added to expose longer‑range information.

**Reviewer Concerns:**

- All reviewers liked that the paper clearly separates over‑squashing into capacity and sensitivity failure modes and that the Neighbor Associative Recall (NAR) benchmark isolates the capacity aspect.
- Reviewers note that the empirical results are strong: gLSTM attains near‑perfect NAR recall up to its memory dimension and achieves state‑of‑the‑art or competitive performance on long‑range real‑world benchmarks (GPP, LRGB, OGBN, heterophilic datasets).
- Two reviewers mention the lack of a formal theoretical analysis of storage capacity as a limitation.
- Reviewers QVwX and mSY2 criticize the original submission for missing strong baselines (graph transformers, rewiring methods), whereas the authors’ rebuttal and the later reviews (e.g., 6qip) note that they have added extensive transformer and rewiring baselines.
- Reviewer ekHn and 6qip question whether the observed improvements stem from the memory component or from the K‑hop aggregation.
- Reviewers 6dpm, QVwX, and ekHn argue that the core architecture is largely an adaptation of existing xLSTM/fast‑weight ideas and thus offers limited technical novelty.
- Reviewers 6dpm, QVwX and mSY2 raise concerns about the quadratic cost of the memory matrix and unclear scalability to >1 M‑node graphs.

**Reviewer Scores:**

Many reviewers decided to raise their scores. The rebuttal addressed the concerns as follows:

- Novelty of the architecture: The authors reply that the vector‑memory coupling (the vector updates the matrix and the matrix is queried to produce the next vector) is a new construct for graph models and not merely a copy of existing xLSTM or fast‑weight modules. They emphasize that this coupling together with graph‑specific recurrence distinguishes gLSTM from prior LSTM/xLSTM work.

- Quadratic cost and scalability: In response to concerns about the memory matrix’s quadratic cost and scalability beyond 1 M nodes, the authors provide runtime and VRAM scaling curves (Appendix D4 Fig 8) and report empirical experiments on large graphs (ogbn‑arxiv with ~$\approx 170 k$ nodes and ogbn‑products with $\approx$ 2.5 M nodes), showing that gLSTM can be trained at that scale.

- Missing strong baselines: The authors acknowledge the original omission and add extensive baselines in the revised version, including graph transformers (GraphGPS, Exphormer, NAGphormer, Polynormer) and rewiring methods (shown in Table 2 and Table 4).

- Attribution of gains (memory vs. K‑hop): To address whether improvements stem from the associative‑memory component or the K‑hop aggregation, the authors present K‑hop ablation studies (Appendix D3 Table 8, Appendix D4 Table 9). These show that K‑hop sometimes harms performance and that gLSTM remains strong without K‑hop on large OGBN datasets, supporting the claim that the increased storage capacity is the primary driver of the observed gains.

---

### Decision · Program_Chairs · 2026-01-26

Accept (Poster)